# Identifiable Generative Models for Missing Not at Random Data Imputation

**Chao Ma**[1,2] *   **Cheng Zhang**[2]
[1]University of Cambridge   [2]Microsoft Research Cambridge
`cm905@cam.ac.uk`
`cheng.zhang@microsoft.com`

## Abstract

Real-world datasets often have missing values associated with complex generative processes, where the cause of the missingness may not be fully observed. This is known as missing not at random (MNAR) data. However, many imputation methods do not take into account the missingness mechanism, resulting in biased imputation values when MNAR data is present. Although there are a few methods that have considered the MNAR scenario, their model's identifiability under MNAR is generally not guaranteed. That is, model parameters can not be uniquely determined even with infinite data samples, hence the imputation results given by such models can still be biased. This issue is especially overlooked by many modern deep generative models. In this work, we fill in this gap by systematically analyzing the identifiability of generative models under MNAR. Furthermore, we propose a practical deep generative model which can provide identifiability guarantees under mild assumptions, for a wide range of MNAR mechanisms. Our method demonstrates a clear advantage for tasks on both synthetic data and multiple real-world scenarios with MNAR data.

## 1   Introduction

Missing data is an obstacle in many data analysis problems, which may seriously compromise the performance of machine learning models, as well as downstream tasks based on these models. Being able to successfully recover/impute missing data in an unbiased way is the key to understanding the structure of real-world data. This requires us to identify the underlying data-generating process, as well as the probabilistic mechanism that decides which data is missing.

In general, there are three types of missing mechanisms [44]. The first type is missing completely at random (MCAR), where the probability of a data entry being missing is independent of both the observed and unobserved data (Figure 1 (a)). In this case, no statistical bias is introduced by MCAR. The second type is missing at random (MAR), which assumes that the missing data mechanism is independent of the value of unobserved data (Figure 1 (b)). Under this assumption, maximum likelihood learning methods without explicit modeling of the missingness mechanism can be applied by marginalizing out the missing variables [3, 25, 28]. However, both MCAR and MAR do not hold in many real-world applications, such as recommender systems [8, 14], healthcare [13], and surveys [51]. For example, in a survey, participants with financial difficulties are more likely to refuse to complete the survey about financial incomes. This is an example of missing not at random (MNAR), where the cause of the missingness (financial income) can be unobserved. In this case, ignoring the missingness mechanism will result in biased imputation, which will jeopardize down-stream tasks.

---

*This work was performed when the authors were (part-time) associated with Microsoft Research, Cambridge

35th Conference on Neural Information Processing Systems (NeurIPS 2021).

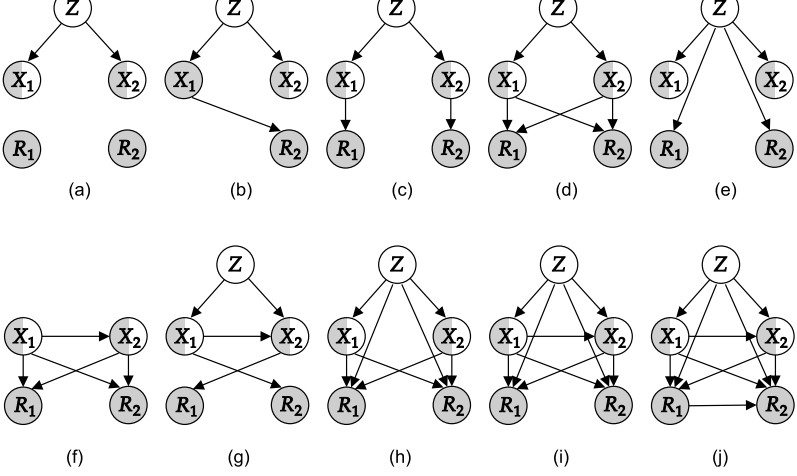

Figure 1: Exemplar missing data situations. **(a)**: MCAR; **(b)**: MAR; **(c)-(i)**: MNAR.

There are few works considering the MNAR setting in scalable missing value imputation. On the one hand, many practical methods for MNAR does not have identifiability guarantees [12, 8, 24]. That is, the parameters can not be uniquely determined, even with access to infinite samples [33, 43]. As a result, missing value imputation based on such parameter estimation could be biased. On the other hand, there are theoretical analyses on the identifiability in certain scenarios [33, 34, 35, 38, 53, 55, 57], but without associated practical algorithms for flexible and scalable settings (such as deep generative models). Moreover, MNAR data have many possible cases (Figure 1) based on different independence assumptions [38], making the discussion of identifiability difficult. This motivates us to fill this gap by extending identifiability results of deep generative models to different missing mechanisms, and provide a scalable practical solution.Our contribution are threefold:

- We provide a theoretical analysis of identifiability for generative models under different MNAR scenarios (Section 3). More specifically, we provide sufficient conditions, under which the ground truth parameters can be uniquely identified via maximum likelihood (ML) learning using observed information [24]. We also demonstrate how the assumptions can be relaxed in the face of real-world datasets. This provides foundation for practical solutions using deep generative models.

- Based on our analysis, we propose a practical algorithm model based on VAEs (Section 4), named GINA (deep generative imputation model for missing not at random). This enables us to apply flexible deep generative models in a principled way, even in the presence of MNAR data.

- We demonstrate the effectiveness and validity of our approach by experimental evaluations (Section 6) on both synthetic data modeling, missing data imputation in real-world datasets, as well as downstream tasks such as active feature selection under missing data.

## 2 Backgrounds

### 2.1 Problem Setting

A critical component to develop model to impute MNAR data is the model identifiablity [18, 43]. We give the definition below:

**Definition 2.1** (Model identifiability). *Assume $p_\theta(X)$ is a distribution of some random variable $X$, $\theta$ is its parameter that takes values in some parameter space $\Omega_\theta$. Then, if $p_\theta(X)$ satisfies $p_{\theta_1}(X) \neq p_{\theta_2}(X) \iff \theta_1 \neq \theta_2, \forall \theta_1, \theta_2 \in \Omega_\theta$, we say that $p_\theta$ is identifiable w.r.t. $\theta$ on $\Omega_\theta$.*

In other words, a model $p_\theta(X)$ is identifiable, if different parameter configurations implies a different probabilistic distributions over the observed variables. With identifiability guarantee, if the model assumption is correct, the true generation process can be recovered. Next, we introduce necessary notations and of missing data, and set up a concrete problem setting.

**Basic Notation.** Similar to the notations introduced by [12, 44], let $X$ be the complete set of variables in the system of interest. We call it *observable variables*. Let $\mathcal{I} = \{1, ..., D\}$ be the *index set* of all observable variables, i.e., $X = \{X_i | i \in \mathcal{I}\}$. Let $X_o$ denote the set of actually *observed variables*, here $O \in \mathcal{I}$ is a index set such that $X_o \subset X$. We call $O$ the *observable pattern*. Similarly, $X_u$ denotes the set of *missing/unobserved variables*, and $X = X_o \bigcup X_u$. Additionally, we use $R$ to denote the missing mask indicator variable, such that $R_i = 1$ indicates $X_i$ is observed, and $R_i = 0$ indicates otherwise. We call a probabilistic distribution $p(X)$ on $X$ the *reference distribution*, that is, the distribution that we would have observed if no missing mechanism is present; and we call the conditional distribution $p(R|X)$ the *missing mechanism*, which decides the probability of each $X_i$ being missing. Then, we can define the marginal distribution of *partially observed variables*, which is given by $\log p(X_o, R) = \log \int_{X_u} p(X_o, X_u, R) dX_u$. Finally, we will use lowercase letters to denote the *realized values* of the corresponding random variable. For example, $(x_o, r) \sim p(X_o, R)$ is the realization/samples of $X_o$ and $R$, and the dimensionality of $x_o$ may vary for each realizations.

**Problem setting.** Suppose that we have a ground truth data generating process, denoted by $p_{\mathcal{D}}(X_o, R)$, from which we can obtain (partially observed) samples $(x_o, r) \sim p_{\mathcal{D}}(X_o, R)$. We also have a model to be optimized, denoted by $p_{(\theta, \lambda)}(X_o, X_u, R)$, where $\theta$ is the parameter of reference distribution $p_\theta(X)$, and $\lambda$ the parameter of missing mechanism $p_\lambda(R|X)$. Our goal can then be described as follows:

- To establish the identifiability of the model $p_{(\theta, \lambda)}(X_o, R)$. That is, we wish to uniquely and correctly identify $\hat{\theta}$, such that $p_{\hat{\theta}}(X) = p_{\mathcal{D}}(X)$, given infinite amount of partially observed data samples from ground truth, $(x_o, r) \sim p_{\mathcal{D}}(X_o, R)$.
- Then, given the identified parameter, we will be able to perform missing data imputation, using $p_{\hat{\theta}}(X_u|X_o)$. If our parameter estimate is unbiased, then our imputation is also unbiased, that is, $p_{\hat{\theta}}(X_u|X_o) = p_{\mathcal{D}}(X_u|X_o)$ for all possible configurations of $X_o$.

## 2.2 Challenges in MNAR imputation

Recall the three types of missing mechanisms: if data is MCAR, $p(R|X) = p(R)$; if if data is MAR, $p(R|X) = p(R|X_o)$; otherwise, we call it MNAR. When missing data is MCAR or MAR, missing mechanism can be ignored when performing maximum likelihood (ML) inference based only on the observed data [44], as:

$$\arg \max_\theta \mathbb{E}_{(x_o, r) \sim p_{\mathcal{D}}(X_o, R)} \log p_\theta(X_o = x_o) = \arg \max_\theta \mathbb{E}_{(x_o, r) \sim p_{\mathcal{D}}(X_o, R)} \log p_\theta(X_o = x_o, R = r)$$

where $\log p(X_o) = \log \int_{X_u} p(X_o, X_u) dX_u$. In practice, ML learning on $X_o$ can done by EM algorithm [3, 24]. However, when missing data is MNAR, the above argument does not hold, and the missing data mechanism cannot be ignored during learning. Consider the representative graphical model example in Figure 1 (d), which has appeared in many context of machine learning. In this graphical model, $X$ is the cause of $R$, and the connections between $X$ and $R$ are fully connected, i.e., each single node in $R$ are caused by the entire set $X$. All nodes in $R$ are conditionally independent from each other given $X$.

Clearly, this is an example of a data generating process with MNAR mechanism. In this case, Rubin proposed to jointly optimize both the reference distribution $p_\theta(X)$ and the missing data mechanism $p_\lambda(R|X)$, by maximizing:

$$\arg \max_{\theta, \lambda} \mathbb{E}_{(x_o, r) \sim p_{\mathcal{D}}(X_o, R)} \left[ \log p_\theta(X_o = x_o) + \log p_\lambda(R = r|X_o = x_o) \right] \tag{1}$$

This factorization is referred as *selection modeling* [12, 24]. There are multiple challenges if we want to Eq. 1 to obtain a practical model that provide unbiased imputation. First, we need model assumption to be consistent with the real-world data generation process, $p_{\mathcal{D}}(X_o, R)$. Given a wide range of possible MNAR scenarios, it is a challenge to design a general model. Secondly, the model need to be identifiable to enable the possibility to learn the underlying process which leads to unbiased imputation.

## 2.3 Variational Autoencoders and its identifiability

Variational auto-encdoers [17, 41, 64] is a flexible deep generative model that is commonly used for estimating densities of $p_{\mathcal{D}}(X)$. It takes the following form:

$$\log p_\theta(X) = \log \int_Z dZ p_\theta(X|Z)p(Z), \tag{2}$$

where $Z$ is some latent variable model with prior $p(Z)$, and $p_\theta(X|Z)$ is given by $p_\theta(X|Z) = \mathcal{N}(f_\theta(Z), \sigma)$, with $f_\theta(\cdot)$ being a neural network parameterized by $\theta$. Generally, VAEs do not have identifiability guarantees w.r.t. $\theta$ [16]. Nevertheless, inspired by the identifiablity of nonlinear ICA, [16] shows that the identifiability of VAE can be established up to equivalence permutation under mild assumptions, if the unconditional prior $p(Z)$ of VAE is replaced by the following the conditionally factorial exponentially family prior,

$$p_{T,\zeta}(Z|U) \propto \prod_{i=1} Q(Z_i) \exp[\sum_{j=1}^K T_{i,j}(Z_i)\zeta_{i,j}(U)], \tag{3}$$

where $U$ is some additional observations (called auxiliary variables), $Q(Z_i)$ is some base measure, $\mathbf{T}_i(U) = (T_{i,1}, ..., T_{i,K})$ the sufficient statistics, and $\boldsymbol{\zeta}_i(U) = (\zeta_{i,1}, ..., \zeta_{i,K})$ the corresponding natural parameters. Then, the new VAE model given by

$$\log p_\theta(X|U) = \log \int_Z dZ p_\theta(X|Z)p_{T,\varsigma}(Z|U) \tag{4}$$

is identifiable (Theorem 1 and 2 of [16], see Appendix G).We call the model (4) the *identifiable VAE*. Unfortunately, this identifiability results for VAE only hold when all variables are fully observed; thus, it cannot be immediately applied to address the challenges of dealing with MNAR data stated in Section 2.2. Next, we will analyze the identifiablity of generative models under general MNAR settings (Section 3), and propose a practical method that can be used in MNAR (Section 4).

# 3 Establishing model identifiability under MNAR

One key issue of training probabilistic models under MNAR missing data is its identifiability. Recall that (Definition 2.1) model identifiability characterize the property that the mapping from parameter $\theta$ to the distribution $p_\theta(X)$ is one-to-one. This is often closely related to maximum likelihood learning. In fact, it is not hard to show that Definition 2.1 is equivalent to the following Definition 3.1:

**Definition 3.1** (Equivalent definition of identifiability). *We say a model $p_\theta(X)$ is identifiable, if:*

$$\arg \max_{\theta \in \Omega_\theta} \mathbb{E}_{x \sim p_{\theta^*}(X)} \log p_\theta(X = x) = \theta^*, \ \ \forall \theta^* \in \Omega_\theta \tag{5}$$

In other words, the "correct" model parameter $\theta^*$ can be identified via maximum likelihood learning (under complete data), and the ML solution is unbiased. Similarly, when MNAR missing mechanism is present, we perform maximum likelihood learning on both $X_o$ and $R$ using Eq. 1. Thus, we need $\log p_{\theta,\lambda}(X_o, R)$ to be identifiable under MNAR, so that we can correctly identify the ground truth data generating process, and achieve unbiased imputation. The identifiability of $\log p_{\theta,\lambda}(X_o, R)$ under MNAR is usually not guaranteed, even in some simplistic settings [33]. In this section, we will give the sufficient conditions for model identifiability under MNAR, and study how these can be relaxed for real-world applications

## 3.1 Sufficient conditions for identifiability under MNAR

In this section, we give sufficient conditions where the model parameters $\theta$ can be uniquely identified by Rubin's objective, Eq. 1. Our aim is to i), find a set of model assumptions, so that it can cover many common scenarios and be flexible for practical interests; and ii), under those conditions, we want to show that its parameters can be uniquely determined by the partial ML solution Eq. 1. As shown in Figure 1, MNAR have many possible difference cases depending on its graphical structures. We want our results to cover every situation.

Instead of doing case by case analysis, we will start our identifiability anaylsis with one fairly general case as the example shown in Figure 1 (h) where the missingness can be caused by other partially

observed variable, by itself (self-masking) or by latent variables. Then, we will discuss how these analysis can be applied to other MNAR scenarios in Section 3.2.

**Data setting D1.** Suppose the ground truth data generation process satisfies the following conditions: all variables $X$ are generated from a shared latent confounder $Z$, and there are no connections among $X$; and the missingness indicator $R$ variable can not be the parent of other variables. A typical example of such distribution is depicted in Figure 1 (h).We further assume that $p_{\mathcal{D}}(X_o, X_u, R)$ has the following parametric form: $p_{\mathcal{D}}(X_o, X_u, R) = \int_Z \prod_d p_{\theta_d^*}(X_d|Z)p(Z)p_{\lambda^*}(R|X, Z)dZ$, where $p_{\lambda^*}(R|X, Z) = \prod_d p_{\lambda_d^*}(R_d|X, Z)$, for some $\theta^*, \lambda^*$.

Then, consider the following model:

**Model assumption A1.** We assume that our model has the same graphical representation, as well as parametric form as **data setting D1**, that is, our model can be written as:

$$p_{\theta,\lambda}(X_o, R) = \int_{X_u, Z} dX_u dZ \prod_d p_{\theta_d}(X_d|Z) \prod_d p_{\lambda_d}(R_d|X, Z)p(Z) \qquad (6)$$

Here, $(\theta, \lambda) \in \Omega$ are learnable parameters that belong to some parameter space $\Omega = \Omega_\theta \times \Omega_\lambda$. Each $\theta$ is the parameter that parameterizes the conditional distribution that connects $X_d$ and $Z$, $p_{\theta_d}(X_d|Z)$. Assume that the ground truth parameter of $p_{\mathcal{D}}$ belongs to the model parameter space, $(\theta^*, \lambda^*) \in \Omega$.

Given such a model, our goal is to correctly identify the ground truth parameter settings given partially observed samples from $p_{\mathcal{D}}(X_o, X_u, R)$. That is, let $(\hat{\theta}, \hat{\lambda}) = \arg\max_{(\theta,\lambda)\in\Omega} \mathrm{E}_{(x_o,r)\sim p_{\mathcal{D}}(X_o,R)} \log p_{(\theta,\lambda)}(X_o = x_o, R = r)$, we would like to achieve $\hat{\theta} = \theta^*$. In order to achieve this, we must make additional assumptions.

**Assumption A2.** Subset identifiability: There exist a partition[2] of $\mathcal{I}$, denoted by $\mathcal{A}_\mathcal{I} = \{O_s\}_{1\leq s\leq S}$, such that: for all $O_s \in \mathcal{A}_\mathcal{I}$, $p_\theta(X_{O_s})$ is identifiable on a subset of parameters $\{\theta_d|d \in O_s\}$.

This assumption basically formalizes the idea of divide and conquer: we partition the whole index set into several smaller subsets $\{O_s\}_{1\leq s\leq S}$, on which each reference distribution $p_\theta(X_{O_s})$ is only responsible for the identifiability on a subset of parameters.

**Assumption A3.** There exists a collection of observable patterns, denote by $\bar{\mathcal{A}}_\mathcal{I} := \{O_l'\}_{1\leq l\leq L}$, such that: 1), $\bar{\mathcal{A}}_\mathcal{I}$ is a cover [2] of $\mathcal{I}$; 2), $p_{\mathcal{D}}(X, R_{O_l'} = 1, R_{\mathcal{I}\setminus O_l'}) > 0$ for all $1 \leq l \leq L$; and 3), for all index $c \in O_l'$, there exists $O_s \in \mathcal{A}_\mathcal{I}$ defined in **A2**, such that $c \in O_s \subset O_l'$.

This assumption is about the strict positivity of the ground truth data generating process, $p_{\mathcal{D}}(X_o, X_u, R)$. Instead of assuming that complete case data are available as in [38], here we assumes we should at least have some observations, $p_{\mathcal{D}}(X, R_o = 1, R_u = 0) > 0$ for $O \in \hat{\mathcal{A}}_\mathcal{I}$, on which $p_\theta(X_o)$ is identifiable.

To summarize, **A1** ensures that our model has the same graphical representation/parametric forms as the ground truth; **A2** $p_\theta(X_o) = \int_{X_u} p_\theta(X_o, X_u)dX_u$ should be at least identifiable for a collection of observable patterns that forms a partition of $\mathcal{I}$; and **Assumption A3** ensures that $p_{\mathcal{D}}(X_o, X_u, R)$ should be positive for certain *important* patterns (i.e., those on which $p_\theta(X_o)$ is identifiable). In Appendix C, we will provide a practical example that satisfies those assumptions. Given these assumptions, we have the following proposition (See Appendix C for proof.):

**Proposition 1** (Sufficient conditions for identifiability under MNAR). *Let $p_{\theta,\lambda}(X_o, X_u, R)$ be a model on the observable variables $X$, and missing pattern $R$, and $p_{\mathcal{D}}(X_o, X_u, R)$ be the ground truth distribution. Assume that they satisfies **Data setting D1, Assumptions A1, A2 and A3**.*

*Let $\Theta = \arg\max_{(\theta,\lambda)\in\Omega} \mathrm{E}_{(x_o,r)\sim p_{\mathcal{D}}(X_o,R)} \log p_{(\theta,\lambda)}(X_o = x_o, R = r)$ be the set of ML solutions of Equation 1. Then, we have $\Theta = \{\theta^*\} \times \Theta_\lambda$. That is, the ground truth model parameter $\theta^*$ can be uniquely identified via (partial) maximum likelihood learning.*

**Missing value imputation as inference.** Given a model $p_{(\theta)}(X_o, X_u)$, the missing data imputation problem can be then formularized by the Bayesian inference problem $p_\theta(X_u|X_o) \propto p_\theta(X_u, X_o)$. If the assumptions of Proposition 1 are satisfied, it enables us to correctly identify the ground truth

---

[2]It can be arbitrary partition in the set theory sense.

reference model parameter, $\theta^*$. Therefore, the imputed values sampled from the posterior $p_{\theta^*}(X_u | X_o)$ will be unbiased, and can be used for down stream decision making tasks.

**Remark:** Note that Proposition 1 can be extended to the case where model identifiability is defined by equivalence classes [16, 53]. See Appendix F for details.

## 3.2 Relaxing "correctness of parametric model" assumption (A1)

In this section, we further extend our previous results to the general MNAR cases including all different examples in Figure 1. In particular, we would like to see the if the same model setting in Section 3.1 can be applied to scenarios where $p_{\mathcal{D}}(X_o, X_u, R)$ and $p_{\theta,\lambda}(X_o, X_u, R)$ might have different parametric forms, or even different graphical representations.

To start with, we would like to point out that the mismatch between $p_{\mathcal{D}}(X_o, X_u, R)$ and the model $p_{\theta,\lambda}(X_o, X_u, R)$ can be, to a certain extend, modeled by the *mappings between spaces of parameters*. Let $\Omega \subset \mathbb{R}^I$ denote the parameter domain of our model, $p_{\theta,\lambda}(X_o, X_u, R)$. Suppose we have a mapping $\Phi : \underline{\Omega} \subset \mathbb{R}^I \mapsto \mathbb{R}^J$, such that $(\theta, \lambda) \in \underline{\Omega} \subset \Omega$ is mapped to another parameter space $(\tau, \gamma) = \Phi(\theta, \lambda) \in \Xi \subset \mathbb{R}^J$ via $\Phi(\cdot)$. Here, $\underline{\Omega}$ is a subset of $\Omega$ on which $\Phi$ is defined. Then, the *re-parameterized* $p_{\theta,\lambda}(X_o, X_u, R)$ on parameter space $\Xi$ can be rewritten as:

$$\tilde{p}_{\tau,\gamma}(X_o, X_u, R) := p_{\Phi^{-1}(\tau,\gamma)}(X_o, X_u, R)$$

Assuming that the inverse mapping $\Phi^{-1}$ exists. Then trivially, if $p_{\theta,\lambda}(X_o, R)$ is identifiable with respect to $\theta$ and $\lambda$, then $\tilde{p}_{\tau,\gamma}(X_o, R)$ should be also identifiable with respect to $\tau$ and $\gamma$:

**Proposition 2.** *Let $\Omega \subset \mathbb{R}^I$ be the parameter domain of the model $p_{\theta,\lambda}(X_o, R)$. Assume that the mapping $\Phi : (\theta, \lambda) \in \underline{\Omega} \subset \mathbb{R}^I \mapsto (\tau, \gamma) \in \Xi \subset \mathbb{R}^J$ is one-to-one on $\underline{\Omega}$ (equivalently, the inverse mapping $\Phi^{-1} : \Xi \mapsto \underline{\Omega}$ is injective, and $\underline{\Omega}$ is its image set). Consider the induced distribution with parameter space $\Xi$, defined as $\tilde{p}_{\tau,\gamma}(X_o, R) := p_{\Phi^{-1}(\tau,\gamma)}(X_o, R)$. Then, $\tilde{p}$ is identifiable w.r.t. $(\tau, \gamma)$, if $p_{\theta,\lambda}(X_o, R)$ is identifiable w.r.t. $\theta$ and $\lambda$.*

Proposition 2 basically shows that if two distributions $p_{\theta,\lambda}(X_o, R)$ and $\tilde{p}_{\tau,\gamma}(X_o, R)$ are related by a mapping $\Phi$ with nice properties, than the identifiability will translate between them. This already covers many scenarios of the data-model mismatch. For example, consider the case where ground truth data generation process satisfies the following assumption:

**Data setting D2** Suppose the ground truth $p_{\mathcal{D}}(X_o, X_u, R)$ satisfies: X are all generated by shared latent confounders Z (as in **D1**), and $R$ cannot be the cause of any other variables as in [38, 56]. Typical examples are given by any of the cases in Fig 1(excluding (j) where $R_1$ is the cause of $R_2$). Furthermore, the ground truth data generating process is given by the parametric form $p_{\mathcal{D}}(X_o, X_u, R) = \tilde{p}_{\tau^*, \gamma^*}(X_o, X_u, R)$, where $\Xi = \Xi_\tau \times \Xi_\gamma$ denotes its parameter space.

Then, for such ground truth data generating process, we can show that we can always find a model in the form of Equation 6, such that there exists some mapping $\Phi$, that can model their relationship:

**Lemma 1.** *Suppose the ground truth data generating process $\tilde{p}_{\tau^*, \gamma^*}(X_o, X_u, R)$ satisfies **setting D2**. Then, there exists a model $p_{\theta,\lambda}(X_o, X_u, R)$, such that: 1), $p_{\theta,\lambda}(X_o, X_u, R)$ can be written in the form of Equation 6 (i.e., **Assumption A1**; and 2), there exists a mapping $\Phi$ as described in Proposition 2, such that $\tilde{p}_{\tau,\gamma}(X_o, R) = p_{\Phi^{-1}(\tau,\gamma)}(X_o, R)$, for all $(\tau, \gamma) \in \Xi$.*

**Model identification under data-model mismatch.** Since we showed the identifiability can be preserved under the parameter space mapping (Proposition 2), next we will prove that if the model $p_{\theta,\lambda}(X_o, X_u, R)$ is trained on partially observed data points sampled from $\tilde{p}_{\tau,\lambda}(X_o, X_u, R)$ that satisfies **data setting D2**, then the ML solution is still unbiased. For this purpose, inspired by Lemma 1, we work with the following additional assumption:

**Model Assumption A4** Let $\tilde{p}_{\tau^*, \gamma^*}(X_o, X_u, R)$ denote our ground truth data generating process that satisfies **data setting D2**. Then, we assume our model $p_{\theta,\lambda}(X_o, X_u, R)$ is the one that satisfies the description given by Lemma 1. That is, its parametric form is given by Equation 6, and there exists a mapping $\Phi$ as described in Proposition 2, such that $\tilde{p}_{\tau,\gamma}(X_o, R) = p_{\Phi^{-1}(\tau,\gamma)}(X_o, R)$.

Then, we have the following proposition:

**Proposition 3** (Sufficient conditions for identifiability under MNAR and data-model mismatch)**.** *Let* $p_{\theta,\lambda}(X_o, X_u, R)$ *be a model on the observable variables* $X$ *and missing pattern* $R$*, and* $p_{\mathcal{D}}(X_o, X_u, R)$ *be the ground truth distribution. Assume that they satisfies **Data setting D2**, **Assumption A2, A3, and A4**. Let* $\Theta = \arg\max_{(\theta,\lambda)\in\underline{\Omega}} \mathrm{E}_{(x_o,r)\sim p_{\mathcal{D}}(X_o,R)} \log p_{(\theta,\lambda)}(X_o = x_o, R = r)$ *be the set of ML solutions of Equation 1. Then, we have* $\Theta = \{\Phi_\tau^{-1}(\tau^*)\} \times \Theta_\lambda$*. Namely, the ground truth model parameter* $\tau^*$ *of* $p_{\mathcal{D}}$ *can be uniquely identified (as* $\Phi(\theta^*)$*) via ML learning.*

**Remark: practical implications** Proposition 3 allows us to deal with the cases where the parameterization of ground truth data generating process and model distribution are related through a set of mappings, $\{\Phi_o\}$. In general, the graphical structure of $p_{\mathcal{D}}(X_o, X_u, R)$ can be any cases in Figure 1 excluding (j). Then, in those cases, we are still able to use a model that corresponds to Equation 6 (Fig 1 (h)) to perform ML learning, provided that our model is flexible enough (**Assumption A4**). This greatly improves the applicability of our identifiability results, and we can build a practical algorithm based on Equation 6 to handle many practical MNAR cases.

## 4 GINA: A Practical Imputation Algorithm for MNAR

In the previous section, we have established the identifiability conditions for models in the form of Equation (6). However, in order to derive a practically useful algorithm, we still need to specify a parametric form of the model, that is both flexible and compatible with our assumptions. In this section, by utilizing the results in Section 3, we propose GINA, a deep generative imputation model for MNAR data (Figure 2). GINA fulfill identifiability assumptions above, and can handle general MNAR case as discussed in section 3.2. The code is released at `https://github.com/microsoft/project-azua`.

**The parametric form of GINA** We use utilize the flexibility of deep generative models to model the data generating process. We assume that the reference model $p_\theta(X)$ is parameterized by an identifiable VAE (see Section 2.3) to satisfy Assumption A2. That is, $p_\theta(X|U) = \int_Z dZ p_\epsilon(X - f(Z)) p(Z|U)$, where $U$ is some fully observed auxiliary inputs. The decoder $p_\epsilon(X - f_\theta(Z))$ is parameterized by a neural network, $f : \mathbb{R}^H \mapsto \mathbb{R}^D$. For convenience, we will drop the input $U$ to $p_\theta(X|U)$, and simply use $p_\theta(X)$ to denote $p_\theta(X|U)$. Finally, for the missing model $p_\lambda(R|X, Z)$, we use a Bernoulli likelihood model, $p_\lambda(R|X, Z) := \prod_d \pi_d(X, Z)^{R_d}(1 - \pi_d(X, Z))^{1-R_d}$, where $\pi_d(X, Z)$ is parameterized by a neural network.

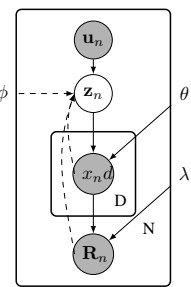

Figure 2: Graphical representations of our GINA.

In Appendix G, we show that GINA fulfill the required assumptions of Proposition 1 and 3. Thus, we can use GINA to identify the ground truth data generating process, and perform missing value imputation under MNAR. The consistency of estimation result is also given in Appendix H.

**Learning and imputation** In practice, the joint likelihood in Equation 1 is intractable. Similar to the approach proposed in [12], we introduce a variational inference network, $q_\phi(Z|X_o)$, which enable us to derive a importance weighted lower bound of $\log p_{\theta,\lambda}(X_o, R)$:

$$\log p_{\theta,\lambda}(X_o, R) \geq \mathcal{L}_K(\theta, \lambda, \phi, X_o, R) := \mathbf{E}_{z^1,\ldots,z^K,x_u^1,\ldots,x_u^K \sim p_\theta(X_u|Z)q_\phi(Z|X_o)} \log \frac{1}{K}\sum_k w_k$$

where $w_k = \frac{p_\lambda(R|X_o, X_u = x_u^k, Z = z^k)p_\theta(X_o, Z = z^k)}{q_\phi(Z = z^k|X_o)}$ is the importance weights. Note that we did not notate the missing pattern $R$ as additional input to $q_\phi$, as this information is already contained in $X_o$. Then, we can optimize the parameters $\theta, \lambda, \phi$ by solving the following optimization problem

$$\theta^*, \lambda^*, \phi^* = \arg\max_{\theta,\lambda,\phi} \mathbb{E}_{(x_o,r)\sim p_{\mathcal{D}}(X_o,R)} \mathcal{L}_K(\theta, \lambda, \phi, X_o = x_o, R = r)$$

Given $\theta^*, \lambda^*, \phi^*$, we can impute missing data by solving the approximate inference problem:

$$p_\theta(X_u|X_o) = \int_Z p_\theta(X_u|Z)p_\theta(Z|X_o)dZ \approx \int_Z p_\theta(X_u|Z)q_\phi(Z|X_o)dZ.$$

Thus, GINA can be used to predict missing data even when the data are MNAR.

# 5 Related works

We mainly review recent works for handling MNAR data. In Appendix A, we provide a brief review of traditional methods that deal with MCAR and MAR.

When the missing data is MNAR, a general framework is to learn a joint model on both observable variables and missing patterns [24], in which a model of missing data is usually assumed [52, 11]. This approach is also widely adopted in imputation tasks. For example, in the field of recommender systems, different probabilistic models are used within such a framework [8, 30, 58, 22, 21]. A similar approach has also been taken in the context of causal approach to imputation [60, 59, 20]. Similar to the use of the missing model, they have used an explicit model of exposure and adopted a causal view, where MNAR is treated as a confounding bias. Apart from these, inverse probability weighting methods are also used to debias the effect of MNAR [48, 58, 29] for imputation.

One issue that is often ignored by many MNAR methods is the model identifiability. Both parametric and non-parametric identifiability under MNAR has been discussed for certain cases ( [34, 33, 35, 57, 55, 53]). For example, [57] proposed the instrumental variable approach to help the identification of MNAR data. [33] investigated the identifiability of normal and normal mixture models, and showed that identifiability for parametric models is highly non-trivial under MNAR. [34] studied conditions for nonparametric identification using shadow variable technique. Despite the resemblance to the auxiliary variable in our approach, [33, 34] mainly considers the supervised learning (multivariate regression) scenario. [38, 37, 50] also discussed a similar topic based on a graphical and causal approach in a non-parametric setting. Although the notion of recoverability has been extensively discussed, their methods do not directly lead to practical imputation algorithms in a scalable setting. On the contrary, our work takes a different approach, in which we handle MNAR with a parametric setting, by dealing with learning and inference in latent variable models. We step aside from the computational burden with the help of recent advances in deep generative models for scalable imputation.

There has been a growing interest in applying deep generative models to missing data imputation. In [28, 25, 40], scalable methods for training VAEs under MAR have been proposed. Similar methods have also been advocated in the context of importance weighted VAEs, multiple imputation [32], and heterogeneous tabular data imputation [40, 26, 27]. Generative adversarial networks (GANs) have also been applied to MCAR data [63, 19]. More recently, deep generative models under MNAR have been studied [12, 5, 6], where different approaches such as selection models [44, 7] and pattern-set mixture models [23] has been combined with partial variational inference for training VAEs. However, without additional assumptions, the model identifiability remains unclear in these approaches, and the posterior distribution of missing data conditioned on observed data might be biased.

# 6 Experiments

We study the empirical performance of the proposed algorithm of Section 4 with both synthetic data (Section 6.1) and two real-world datasets with music recommendation (Section 6.2) and personalized education (Section 6.3) . The experimental setting details can be found in Appendix B.

## 6.1 Synthetic MNAR dataset

We first consider 3D synthetic MNAR datasets. We generate three synthetic datasets with nonlinear data generation process (shown in Appendix B.1). For all datasets, $X_1, X_2, X_3$ are generated via the latent variables, $Z_1, Z_2, Z_3$ ,where $X_1$ are fully observed and $X_2$ and $X_3$ are MNAR. For dataset A, we apply *self-masking*(similar to Figure 1(c)): $X_i$ will be missing if $X_i > 0$. For datasets B and C, we apply *latent-dependent self-masking*: $X_i$ will be missing, if $g(X_i, Z_1, Z_2, Z_3) > 0$, where $g$ is a linear mapping whose coefficients are randomly chosen.

We train GINA and baseline models with partially observed data. Then, we use the trained models to generate random samples. By comparing the generated samples with the ground truth data density, we can evaluate whether $p_{\mathcal{D}}(X)$ is correctly identified. Results are visualized in Figure 3. In addition, we show the imputation results in Appendix J. Across three datasets, PVAE performs poorly, as it does not account for the MNAR mechanism. Not-MIWAE performs better than PVAE, as it is able to generate samples that are closer to the mode. However, it is still biased more towards the observed

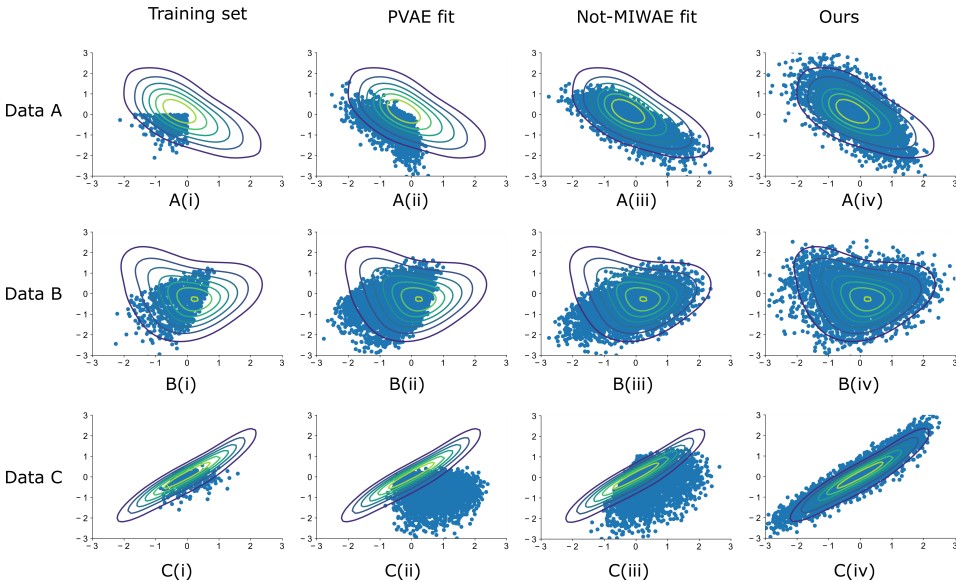

Figure 3: Visualization of generated $X_2$ and $X_3$ from synthetic experiment. **Row-wise (A-C)** plots for dataset A, B, and C, respectively; **Column-wise (i-iv):** training set (only displays fully observed samples), PVAE samples, Not-MIWAE samples, and GINA samples, respectively. **Contour plot**: kernel density estimate of ground truth density of complete data;

values. On the other hand, GINA is much more aligned to ground truth, and is able to recover the ground truth from partially observed data. This experiment showed the clear advantage of our method under different MNAR situations.

## 6.2 Recommender dataset imputation with random test set

| Method | Test MSE |
|---|---|
| *Matrix Factorization Methods* | |
| PMF | 1.401 |
| IPW-PMF | 1.375 |
| Deconfounded-PMF | 1.329 |
| PMF-MNAR | 1.483 |
| PMF-MAR | 1.480 |
| *VAE-based models* | |
| PVAE | 1.259±0.003 |
| PVAE w/o IW | 1.261±0.004 |
| Not-MIWAE | 1.078±0.000 |
| **GINA** | **1.052±0.002** |
| *Others* | |
| CPTv-MNAR | 1.056 |
| Logitvd-MNAR | 1.141 |
| AutoRec | 1.199 |
| Oracle-test | 1.057 |

Table 1: Test MSE on Yahoo! R3

Next, we apply our models to recommendation systems on Yahoo! R3 dataset [30, 60] for user-song ratings which is designed to evaluate MNAR imputation. It contains an MNAR training set of more than 300K self-selected ratings from 15,400 users on 1,000 songs, and an MCAR test set of randomly selected ratings from 5,400 users on 10 random songs. We train all models on the MNAR training set, and evaluate on MCAR test set. This is repeated 10 times with different random seeds. Both the missing model for GINA ($p(R|X, Z)$) and Not-MIWAE ($p(R|X)$) are parameterized by linear neural nets with Bernoulli likelihood functions. The decoders for GINA, PVAE, and Not-MIWAE uses Gaussian likelihoods with the same network structure. See Appendix B for implementation details and network structures.

We compare to the following baselines: 1), probabilistic matrix factorization (PMF) [36]; 2), inverse probability weighted PMF [48]; 3), Deconfounded PMF [60]; 4), PMF with MNAR/MAR data [8]; 5), CPTv and Logitv models for MNAR rating [30]; 6), Oracle [8]: predicts ratings based on their marginal distribution in the test set; and 7) AutoRec [49]: Autoencoders that ignores missing data.

Results are shown in Table 1. Our method (GINA) gives the best performance among all methods. Also, VAE-based methods are consistently better than PMF-based methods, and MNAR-based models consistently outperform their MAR versions. More importantly, among VAE-based models, our GINA outperforms its non-identifiable counterpart (Not-MIWAE), and MAR counterpart (PVAE), where both models can not generate unbiased imputation.

### 6.3 Missing data imputation and active question selection on Eedi education dataset

Finally, we apply our methods to the Eedi education dataset [61], one of the largest real-world education response datasets. We consider the Eedi competition task 3 dataset, which contains over 1 million responses from 4918 students to 948 multiple-choice diagnostic questions. Each diagnostic question is a multiple-choice question. We consider predicting whether a student answers a question correctly or not. Over 70% of the entries are missing. The dataset also contains student metadata which we use as the auxiliary variables. In this experiment, we randomly split the data in a 90% train/ 10% test/ 10% validation ratio, and train our models on the response outcome data.

We evaluate our model on two tasks. Firstly, we perform missing data imputation, where different methods perform imputation over the test set. As opposed to Yahoo! R3 dataset, now the test set is MNAR, thus we use the evaluation method suggested by [60], where we evaluate the average per-question MSE For each question, over all students with non-empty response. Then, the MSEs of all questions averaged. We call this metric the debiased MSE. While regular MSE might be biased toward questions with more responses, the debiased MSE treats all questions equally, and can avoid selection bias to a certain degree. We report results for 10 repeats in the first column in Table 2. We can see that our proposed GINA achieves significantly improved results comparing to the baselines.

Secondly, we evaluate personalized education through active question selection [28] on the test set which is task 4 from this competition dataset. The procedure is as follows: for each student in the test set, at each step, the trained generative models are used to decide which is the most informative missing question to collect next. This is done by maximizing the information reward as in [28] (see Appendix I for details). Since at each step, different students might collect different questions, there isn't a simple way to debias the predictive MSE as in the imputation task. Alternatively, we evaluate each method with the help of *question meta data* (difficulty level, which is a scalar). Intuitively, when the student response to the previously collected question is correct, we expect the next diagnostic question which has higher difficulty levels, and vice versa. Thus, we can evaluate the mean level change after correct/incorrect responses, and expect them to have significant differences. We also perform t-test between the level changes after incorrect/correct responses and report the p-value.

Table 2: Performance on Eedi education dataset (with standard errors)

| Method | Debiased MSE | Level change (correct) | Level change (incorrect) | p-value |
|---|---|---|---|---|
| PVAE | $0.194\pm0.001$ | $0.131\pm0.138$ | $-0.101\pm0.160$ | $0.514$ |
| Not-MIWAE | $0.192\pm0.000$ | $0.062\pm0.142$ | $-0.073\pm0.179$ | $0.561$ |
| GINA | $\mathbf{0.188\pm0.001}$ | $\mathbf{0.945\pm0.151}$ | $\mathbf{-0.353\pm0.189}$ | $\mathbf{1.01\times10^{-7}}$ |

We can see in Table 2, GINA is the only method that reports a significant p-value ($<0.05$) between the level changes of next collected questions after incorrect/correct responses which are desired. This further indicates that our proposed GINA predicts the unobserved answer with the desired behavior.

## 7 Conclusion

In this paper, we provide a analysis of identifiability for generative models under MNAR, and studies sufficient conditions of identifiability under different scenarios. We provide sufficient conditions under which the model parameters can be uniquely identified, via joint maximum likelihood learning on $X_o$ and $R$. Therefore, the learned model can be used to perform unbiased missing data imputation. We proposed a practical algorithm based on VAEs, which enables us to apply flexible generative models that is able to handle missing data in a principled way. The main limitation of our proposed practical algorithm is the need for auxiliary variables $U$, which is inherited from identifiable VAE models [16]. In practice, they may not be always available. For future work, we will investigate how to address such limitation, and how to extend to more complicated scenarios.

## Acknowledgements and Disclosure of Funding

We thank Martin Kukla, Angus Lamb, Yingzhen Li, Dawen Liang, Chang Liu, Ruibo Tu, Yordan Zaykov (in alphabetical order) for helpful discussions and implementation support.

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
