# A Traditional methods for handling missing data

Methods for handling missing data has been extensively studied in the past few decades. Those methods can be roughly classified into two categories: complete case analysis (CCA) based, and imputation based methods. CCA based methods, such as listwise deletion [1] and pairwise deletion [31] focuses on deleting data instances that contains missing entries, and keeping those that are complete. Listwise/pairwise deletion methods are known to be unbiased under MCAR, and will be biased under MAR/MNAR. On the contrary, imputation based methods tries to replace missing values by imputed/predicted values. One popular imputation technique is called single imputation, where only produce one single set of imputed values for each data instance. Standard techniques of single imputation include mean/zero imputation, regression-based imputation [1], no- parametric methods [15, 54]. As opposed to single imputation, the multiple imputation (MI) methods such as MICE [62], was first proposed by Rubin [45, 46, 9, 39] is essentially a simulation-based methods that returns multiple imputation values for subsequent statistical analysis. Unlike single imputation, the standard errors of estimated parameters produced with MI is known to be unbiased [47]. Apart from MI, there exists other methods such as full information maximum likelihood [2, 4] and inverse probability weighting [42, 10], which can be directly applied to MAR without introducing additional bias. However, these methods assumes a MAR missing data mechanism, and cannot be directly applied to MNAR without introducing bias.

# B Implementation details

We first introduce the general settings of GINA and other baselines. Our model (GINA) is based on the practical algorithm in Section 4. By default, we will set the auxiliary variable $U$ to be some fully observed meta feature (if there's any) or the missing mask pattern (if the dataset does not have a fully observed meta feature). The most important baselines are as follows: 1), Partial VAE (PVAE) [28]: a VAE model with slightly modified ELBO objective, specifically designed for MAR data; and 2), Not-MIWAE [12], a VAE model for MNAR data trained by jointly maximizing the likelihood on both the partially observed data and the missing pattern. As opposed to our model, the latent priors $p(Z)$ for both PVAE and Not-MIWAE are parameterized by a standard normal distribution, hence no auxiliary variables are used. Also, note that the graphical model of Not-MIWAE is described by Fig 1 (d), and does not handle the scenarios where the ground truth data distribution follows other graphs like Fig 1 (g). Finally, the inference model $q(Z|X)$ for the underlying VAEs is set to be diagonal Gaussian distributions whose mean and variance are parameterized by neural nets as in standard VAEs [17] (with missing values replaced by zeros[40, 12, 32]), or a permutation invariant set function proposed in [28]. See Appendix B for more implementation details for each tasks.

## B.1 Synthetic dataset implementation details

**Data generation**  The ground truth data generating process is given by $Z_1, Z_2, Z_3 \sim \mathcal{N}(0,1)$, $X_1 = h_w(Z_1, Z_2, Z_3) + \epsilon_1$, $X_2 = f_{\theta_1}(X_1, Z_1, Z_2, Z_3) + \epsilon_2$, $X_3 = f_{\theta_2}(X_1, X_2, Z_1, Z_2, Z_3) + \epsilon_3$ where $h_w$ is a linear mapping with coefficients $w$, $f$ is some non-linear mapping whose functional form is given by Appendix B, $\theta_1$ & $\theta_2$ are two different sets of parameters for $f$, and $\epsilon_1, \epsilon_2, \epsilon_3$ are observational noise variables with mean 0, variance 0.01. We randomly sample three different sets of parameters, and generate the corresponding datasets (Figure 3), namely dataset A, B, and C. Each dataset consists of 2000 samples. Then, we apply different missing mechanisms for each dataset. For all datasets, we assume that $X_1$ are fully observed and $X_2$ and $X_3$ are MNAR. , and missing mechanism will be only applied to $X_2$ and $X_3$. Finally, all observable variables are standardized.

**Remark**  Note that in Dataset A, the ground truth missing mechanism does not depend on the latent variable model. Therefore, in this case, the not-MIWAE model does not have model-misspecification problem, hence any less satisfying performance is due to non-identifiability.

**Network structure and training**  We use 5 dimensional latent space with fully factorized standard normal priors. The decoder part $p_\theta(X|Z)$ uses a 5-10-$D$ structure, where $D = 3$ in our case. For inference net, we use a zero imputing [28] with structure $2D$-10-10-5, that maps the concatenation of observed data (with missing data filled with zero) and mask variable $R$ into distributional parameters of the latent space. For the factorized prior $p(Z|U)$ of the i-VAE component of GINA, we used

a linear network with one auxiliary input (which is set to be fully observed dimension, $X_1$). The missing model $p_\lambda(R|X)$ for GINA and i-NotMIWAE is a single layer neural network with 10 hidden units. All neural networks use Tanh activations (except for output layer, where no activation function is used). All baselines uses importance weighted VAE objective with 5 importance samples. The observational noise for continuous variables are fixed to $\log \sigma = -2$. All methods are trained with Adam optimizer with batchsize with 100, and learning rate 0.001 for 20k epochs.

## B.2 Yahoo!R3 experiment implementation details

Before training, all user ratings are scaled to be between 0 and 1 (such scaling will be reverted during evaluation). For all baselines, we use Gaussian likelihood with variance of 0.02. We use 20 dimensional latent space, and the decoder $p_\theta(X|Z)$ uses a 20-10-$D$ structure. We use Tanh activation function for the decoder (except for output layer, where no activation function is used). For inference net, we uses the point net structure proposed in [28], we use 20 dimensional feature mapping $h$ parameterized by a single layer neural network and 20 dimensional ID vectors for each variable. The symmetric operator is set to be the summation operator. The missing model $p_\lambda(R = 1|X)$ for GINA and i-NotMIWAE is parameterized by linear neural network. All methods are trained with 400 epochs with batchsize 100.

## B.3 Eedi dataset experiment implementation details

Since Eedi dataset is a binary matrix with 1/0 indicating that the student response is correct/incorrect, we use Bernoulli likelihood for decoder $p_\theta(X|Z)$. For We use 50 dimensional latent space, and the decoder $p_\theta(X|Z)$ uses a 50-20-50-$D$ structure. Such structure is chosen using the validation set using grid search. We use ReLU activation function for the decoder (except for output layer, where no activation function is used). For inference net, we uses the point net structure that were used in Yahoo!R3 dataset. Here, the difference is that we we use 50 dimensional feature mapping $h$ parameterized by a single layer neural network and 10 dimensional ID vectors for each variable. All methods are trained with 1k epochs with batchsize 100. A trick that we used for both not-MIWAE and GINA to improve the imputation performance, is to turn down the weight of the likelihood term for $p_\lambda(R|X)$, by multiplying a factor of $\beta = 0.5$. This is due to that majority of the student response matrix is missing, the $p_\lambda(R|X)$ will most likely dominate the training, hence the learning algorithm will prefer more about learning the models that explains the missing mechanism better, over the models that explains the observable variables $X$ better.

## C  Proof for Proposition 1

**Proof**  : First, we show that $p_{\theta,\lambda}(X_{O'_l}, R)$ is *partially identifiable* (i.e., identifiable on subset of parameters) on $\{\theta_d\}_{d \in O'_l}$ for $\forall O'_l \in \bar{\mathcal{A}}_\mathcal{I}$. We prove this by contradiction. Suppose there exists two different set of parameters $(\theta^1, \lambda^1)$ $(\theta^2, \lambda^2)$, such that there exits at least one index $c \in O'_l$ for some $l$, such that $\theta_c^1 \neq \theta_c^2$, and $p_{(\theta^1, \lambda^1)}(X_{O'_l}, R) = p_{(\theta^2, \lambda^2)}(X_{O'_l}, R)$. That is, $p(X_{O'_l}, R)$ is not identifiable on $\{\theta_d\}_{d \in O'_l}$.

According to **Assumption A3**, there exists $O_s \in \mathcal{A}_\mathcal{I}$, such that $c \in O_s \subset O'_l$. Then, consider the marginal

$$p_\theta(X_{O_s}) = \int_{Z, R, X_{\setminus O_s}} dZ \prod_{d \in O_s} p_{\theta_d}(X_d|Z) p_\lambda(R|X, Z) p(Z) = p_{\theta_{d \in O_s}}(X_{O_s})$$

.  Since $p_{(\theta^1, \lambda^1)}(X_{O'_l}, R) = p_{(\theta^2, \lambda^2)}(X_{O'_l}, R)$, we have $p_{(\theta^1_{O_s})}(X_{O_s}) = p_{(\theta^2_{O_s})}(X_{O_s})$ (the joint uniquely determines marginals). However, this contradicts with our **Assumption A2** that $p_{\theta_{O_s}}(X_{O_s})$ is identifiable: this identifiability assumption implies that we should have $p_{(\theta^1_{O_s})}(X_{O_s}) \neq p_{(\theta^2_{O_s})}(X_{O_s})$. Therefore, by contradiction, we have $p(X_{O'_l}, R)$ is partially identifiable on $\{\theta_d\}_{d \in O'_l}$ for $\forall O'_l \subset \bar{\mathcal{A}}_\mathcal{I}$.

Then, we proceed to prove that the ground truth parameter $\theta^*$ can be uniquely identified via ML learning. Based on our **Assumption A1**, upon optimal ML solution,

$$\Theta_{ML} = \arg \max_{(\theta, \lambda) \in \Omega} \mathrm{E}_{(x_o, r) \sim p_\mathcal{D}(X_o, R)} \log p_{(\theta, \lambda)}(X_o = x_o, R = r)$$

, we have the following identity:
$$p_{(\theta_{ML}, \lambda_{ML})}(X_o, R) = p_{(\theta^*, \lambda^*)}(X_o, R)$$
holds for all $(\theta_{ML}, \lambda_{ML}) \in \Theta_{ML}$, and all $\forall O \subset \mathcal{I}$ that satisfies $p(X_o, X_u, R_o = 1, R_u = 0) > 0$.
Note also that:

$$p_{(\theta_{ML}, \lambda_{ML})}(X_o, R) = \int_{Z, X_{\mathcal{I} \setminus O}} dZ \prod_d p_{\theta_d^{ML}}(X_d | Z) p_{\lambda^{ML}}(R | X) p(Z)$$

, which depends on both $\theta_o$ and $\lambda$. Since we have already shown that $p_{(\theta, \lambda)}(X_{O'_l}, R)$ are partially identifiable on $\{\theta_d\}_{d \in O'_l}$ for $\forall O'_l \subset \bar{\mathcal{A}}_{\mathcal{I}}$ and according to **Assumption A3**, $p_{\mathcal{D}}(X_o, X_u, R_{O'_l} = 1, R_{\mathcal{I} \setminus O'_l} = 0) > 0$. Therefore, upon optimal solution

, we have that
$$\{\theta_d = \theta_d^*\}_{d \in O'_l}$$

holds for all $\forall O'_l \subset \bar{\mathcal{A}}_{\mathcal{I}}$. Since we have assumed that $\bigcup_{O'_l \in \bar{\mathcal{A}}_{\mathcal{I}}} X_{O'_l} = \mathcal{I}$ in **Assumption 3** (i.e.,$\bar{\mathcal{A}}_{\mathcal{I}}$ is a cover of $\mathcal{I}$ ), this guarantees that

$$\theta_d^{ML} = \theta_d^*$$
for all $d$. In other words, we are able to uniquely identify $\theta^*$ from observed data, therefore
$$\Theta = \{\theta^*\} \times \Theta_\lambda$$
. $\qquad\qquad\qquad\qquad\qquad\qquad\qquad\qquad\qquad\qquad\qquad\qquad\qquad\qquad\qquad\qquad\square$

**Remark (examples).** To better illustrated the implication of the proposition, we provide an example that satisfies the assumptions of Proposition 1. One example is the self-masking one-way ANOVA [24], which contains $I$ observable variables, $X = \{X_1, ..., X_I\}$, generated according to
$$X_i | Z_i \sim \mathcal{N}(X_i; Z_i, \sigma^2), \quad i = 1, ..., I,$$
where $\sigma^2$ is some known observational noise variance, $Z = \{Z_1, ..., Z_I\}$ are latent variables generated by
$$Z_i \sim \mathcal{N}(Z_i; \mu_i, \varsigma^2).$$
Hence, the learnable parameter $\theta$ is given by $\theta = \{\mu_1, ..., \mu_I, \varsigma^2\}$. Then, assume that each observable variable $X_i$ is missing MNAR according to the following mechanism:
$$p(R_i = 1 | X, Z) = \text{Sigmoid}(\lambda_1 Z_i + \lambda_0),$$
where $\lambda = \{\lambda_0, \lambda_1\}$ is the set of learnable parameters for missing mechanism. Now, we verify that $p_{\theta, \lambda}(X, Z)$ satisfies the assumptions of proposition 1:

- **Assumption A2**. By taking $\mathcal{A}_{\mathcal{I}} = \{O_s\}_{1 \le s \le I}$, where $O_s = \{s\}$, we have $p_\theta(X_s) = \mathcal{N}(X_s; \mu_s, \sigma^2 + \varsigma^2)$, which is identifiable over the subset of parameters $\theta_s = \{\mu_s, \varsigma^2\}$. Note that the partition $\mathcal{A}_{\mathcal{I}} = \{O_s\}_{1 \le s \le I}$ is not unique; in fact, since $p_\theta(X_{O_s}) = \prod_{i \in O_s} \mathcal{N}(X_i; \mu_i, \sigma^2 + \varsigma^2)$ is identifiable on $\{\mu_s | s \in O_s\} \bigcup \{\varsigma^2\}$ for all non-empty $O_s \subset \mathcal{I}$, any partition $\mathcal{A}_{\mathcal{I}} = \{O_s\}_{1 \le s \le I}$ will satisfy **Assumption A2**.
- **Assumption A3**. Since $p_\theta(X)$ and $p_\lambda(R | X, Z)$ are strictly positive for all possible settings of $X, R, \theta$ and $\lambda$, **Assumption A3** is trivially satisfied.

# D Proof of Proposition 2

**Proof** Let $(\tau_1, \gamma_1)$ and $(\tau_2, \gamma_2)$ be two different parameters in $\Xi$. Then, we have
$$\tilde{p}_{\tau_1, \gamma_1}(X_o, R)$$
$$= p_{\Phi^{-1}(\tau_1, \gamma_1)}(X_o, R)$$
$$\neq p_{\Phi^{-1}(\tau_2, \gamma_2)}(X_o, R)$$
$$= \tilde{p}_{\tau_2, \gamma_2}(X_o, R)$$
where the third line is due to the fact that $\Phi^{-1}$ is injective and $p_{\theta, \lambda}(X_o, R)$ is identifiable with respect to $\theta$ and $\lambda$. $\qquad\qquad\qquad\qquad\qquad\qquad\qquad\qquad\qquad\qquad\qquad\qquad\square$

# E  Relaxing Assumption A1

## E.1  Proof of Lemma 1

**Lemma 1.** *Suppose the ground truth data generating process $\tilde{p}_{\tau^*,\gamma^*}(X_o, X_u, R)$ satisfies **setting D2**. Then, there exists a model $p_{\theta,\lambda}(X_o, X_u, R)$, such that: 1), $p_{\theta,\lambda}(X_o, X_u, R)$ can be written in the form of Equation 6 (i.e., **Assumption A1**; and 2), there exists a mapping $\Phi$ as described in Proposition 2, such that $\tilde{p}_{\tau,\gamma}(X_o, R) = p_{\Phi^{-1}(\tau,\gamma)}(X_o, R)$, for all $(\tau, \gamma) \in \Xi$. Additionally, such $\Phi$ is decoupled, i.e., $\Phi(\theta, \lambda) = (\Phi_\theta(\theta), \Phi_\lambda(\lambda))$.*

**Proof:** [3]

**Case 1 (connections among $X$):** Suppose the ground truth data generating process $p_{\mathcal{D}}(X, R) = \tilde{p}_{\tau^*,\gamma^*}(X_o, X_u, R)$ is given by Figure 1 (i). That is, $p_{\mathcal{D}}(X, R) = \tilde{p}_\gamma(X|R) \int_Z \prod_i \tilde{p}_{\tau_i}(X_i|Z, pa(X_i) \bigcap X) p(Z) dZ$. Without loss of generality, assume that probabilistic distributions $\tilde{p}_{\tau_i}(X_i|Z, pa(X_i) \bigcap X)$ takes the form as $\tilde{p}_{\tau_i}(X_i|Z, pa(X_i) \bigcap X) = \int_{\epsilon_i} \delta(X_i - f_i^{\lambda_i}(\epsilon_i, pa(X_i) \bigcap X, Z)) p(\epsilon_i) d\epsilon_i$. Therefore, we have

$$
\tilde{p}_\tau(X)
$$
$$
= \int_Z \prod_i \tilde{p}_{\tau_i}(X_i|Z, pa(X_i) \bigcap X) p(Z) dZ
$$
$$
= \int_z \left[ \prod_{\{i|N(X_i) \bigcap X \neq \emptyset\}} \int_{\epsilon_i} d\epsilon_i \delta(X_i - f_i^{\lambda_i}(\epsilon_i, pa(X_i) \bigcap X, Z)) p(\epsilon_i) \right]
$$
$$
\left[ \prod_{\{j|N(X_j) \bigcap X = \emptyset\}} p(X_j|Z) \right] p(Z) dZ
$$
$$
= \int_{z, \{i|N(X_i) \bigcap X \neq \emptyset\}} \left[ \prod_{\{i|N(X_i) \bigcap X \neq \emptyset\}} \delta(X_i - f_i^{\lambda_i}(\epsilon_i, pa(X_i) \bigcap X, Z)) p(\epsilon_i) \right]
$$
$$
\left[ \prod_{\{j|N(X_j) \bigcap X = \emptyset\}} p(X_j|Z) \right] p(Z) dZ
$$

Apparently, there exists a set of function $\{g_i(\cdot)|N(X_i) \bigcap X \neq \emptyset\}$, such that:

$$
\int_{z, \{i|N(X_i) \bigcap X \neq \emptyset\}} \left[ \prod_{\{i|N(X_i) \bigcap X \neq \emptyset\}} \delta(X_i - f_i^{\lambda_i}(\epsilon_i, pa(X_i) \bigcap X, Z)) p(\epsilon_i) \right]
$$
$$
\left[ \prod_{\{j|N(X_j) \bigcap X = \emptyset\}} p(X_j|Z) \right] p(Z) dZ
$$
$$
= \int_{z, \{i|N(X_i) \bigcap X \neq \emptyset\}} \left[ \prod_{\{i|N(X_i) \bigcap X \neq \emptyset\}} \delta(X_i - g_i(\epsilon_i, anc_\epsilon(i), Z)) p(\epsilon_i) \right]
$$
$$
\left[ \prod_{\{j|N(X_j) \bigcap X = \emptyset\}} p(X_j|Z) \right] p(Z) dZ
$$

---

[3]We mainly consider the case where all variables are continuous. Discrete variables will complicate the discussion, but will not change the conclusion.

Where $anc_\epsilon(i)$ is the shorthand for

$$\{\epsilon_k | X_k \in ancX_i \bigcap Z, 1 \le k \le D\}$$

Note that, the graphical model of the new parameterization,

$$p(X) = \int_{z, \{i | N(X_i) \bigcap X \ne \emptyset\}} \left[ \prod_{\{i | N(X_i) \bigcap X \ne \emptyset\}} \delta(X_i - g_i(\epsilon_i, anc_\epsilon(i), Z))p(\epsilon_i) \right]$$

$$\left[ \prod_{\{j | N(X_j) \bigcap X = \emptyset\}} p(X_j | Z) \right] p(Z)dZ$$

has a new aggregated latent space, $\{Z, \{\epsilon_i | 1 \le i \le D\}\}$. That is, for each $X_i$ that has non empty neighbour in $X$, a new latent variable will be created. With this new latent space, the connections among $X$ can be decoupled, and the new graphical structure of $p(X, R)$ corresponds to Figure 1 (h).

The mapping $\Phi$ that connects $\tilde{p}_{\tau_i}(X, R)$ and $p(X, R)$ can now be defined as identity mapping, since no new parameters are introduced/removed when reparameterizing $\tilde{p}_{\tau_i}(X, R)$ into $p(X, R)$. Hence, the two requirements of Lemma 1 are fulfilled.

**Case 2(subgraph):** Next, consider the case that the ground truth data generating process $p_\mathcal{D}(X, R) = \tilde{p}_{\tau^*, \gamma^*}(X_o, X_u, R)$ is given by one of the Figure 1 (a)-(g). That is, it is a subgraph of Figure 1 (h). Without loss of generality, assume that $\tilde{p}_{\gamma_i}(R_i = 1 | pa(R_i)) = \text{logit}^{-1}(f_{\gamma_i}(pa(R_i)))$, and $pa(R_i) \subsetneq \{X, Z\}$; in other words, certain connections from $\{X, Z\}$ to $R_i$ is missing. Consider the model distribution parameterized by $p(R_i = 1 | X, Z) = \text{logit}^{-1}(f_{\gamma_i}(pa(R_i)) + g_{\theta_i}(\{X, Z\} \setminus pa(R_i)))$, satisfying $g_{\theta_i = 0}(\cdot) \equiv 0$. Therefore, the mapping $\Phi^{-1}$ is given as $\Phi^{-1}(\gamma_i) := (\gamma_i, \theta_i = 0)$. Apparently, $\Phi^{-1}$ is injective, hence satisfying the requirement of Proposition 2.

$\square$

### E.2 Proof for Proposition 3

**Proposition 3** (Sufficient conditions for identifiability under MNAR and data-model mismatch). *Let $p_{\theta, \lambda}(X_o, X_u, R)$ be a model on the observable variables $X$ and missing pattern $R$, and $p_\mathcal{D}(X_o, X_u, R)$ be the ground truth distribution. Assume that they satisfies **Data setting D2, Assumption A2, A3, and A4**. Let $\Theta = \arg\max_{(\theta, \lambda) \in \Omega} E_{(x_o, r) \sim p_\mathcal{D}(X_o, R)} \log p_{(\theta, \lambda)}(X_o = x_o, R = r)$ be the set of ML solutions of Equation 1. Then, we have $\Theta = \{\Phi_\tau^{-1}(\tau^*)\} \times \Theta_\lambda$. Namely, the ground truth model parameter $\tau^*$ of $p_\mathcal{D}$ can be uniquely identified (as $\Phi(\theta^*)$) via ML learning.*

**Proof** : First, it s not hard to show that $p_{\theta, \lambda}(X_{O_l'}, R)$ is partially identifiable on $\{\theta_d\}_{d \in O_l'}$ for $\forall O_l' \in \bar{\mathcal{A}}_\mathcal{I}$. This has been shown in the proof of Proposition 1, and we will not repeat this proof again.

Next, given data setting **D2** and **Assumption A4**, define

$$\Theta_{ML} = \arg\max_{(\theta, \lambda) \in \Omega} E_{(x_o, r) \sim p_\mathcal{D}(X_o, R)} \log p_{(\theta, \lambda)}(X_o = x_o, R = r)$$

, then we have:

$$p_{(\theta_{ML}, \lambda_{ML})}(X_o, R) = p_{\Phi^{-1}(\tau^*, \gamma^*)}(X_o, R)$$

holds for all $(\theta_{ML}, \lambda_{ML}) \in \Theta_{ML}$, and all $\forall O \subset \mathcal{I}$ that satisfies $p(X_o, X_u, R_o = 1, R_u = 0) > 0$.

Since $p_{(\theta, \lambda)}(X_{O_l'}, R)$ are partially identifiable on $\{\theta_d\}_{d \in O_l'}$ for $\forall O_l' \subset \bar{\mathcal{A}}_\mathcal{I}$ and according to **Assumption A3**, $p_\mathcal{D}(X_o, X_u, R_{O_l'} = 1, R_{\mathcal{I} \setminus O_l'} = 0) > 0$. Therefore,

$$\{\theta_d = \Phi_\theta^{-1}(\tau^*, \gamma^*)_d\}_{d \in O_l'}$$

must be true for all $\forall O_l' \subset \bar{\mathcal{A}}_\mathcal{I}$, where $\Phi_\theta^{-1}(\tau^*, \gamma^*)$ denotes the components of $\Phi^{-1}(\tau^*, \gamma^*)$ that corresponds to the entries of $\theta$. Since we have assumed that $\bigcup_{O_l' \in \bar{\mathcal{A}}_\mathcal{I}} X_{O_l'} = \mathcal{I}$ in **Assumption 3** (i.e., $\bar{\mathcal{A}}_\mathcal{I}$ is a cover of $\mathcal{I}$ ), this guarantees that

$$\theta_d^{ML} = \Phi_\theta^{-1}(\tau^*, \gamma^*)_d$$

for all $d$. In other words, we are able to uniquely identify $\theta^*$ from observed data, therefore

$$\Theta = \{\Phi_\theta^{-1}(\tau^*, \gamma^*)\} \times \Theta_\lambda$$

.

Finally, according to **Assumption 4** and the proof of Lemma 1, $\Phi$ is decoupled as $(\Phi_\theta(\theta), \Phi_\lambda(\lambda))$. Therefore, we can write $\Theta = \{\Phi^{-1}(\tau^*)\} \times \Theta_\lambda$. That is, the ground truth model parameter $\tau^*$ of $p_\mathcal{D}$ can be uniquely identified (as $\Phi(\theta^*)$).

$\square$

## F    Identifiability based on equivalence classes

In this section, we introduce the notion of identifiability based on equivalence classes. Let $\sim$ be an equivalence relation on a parameter space $\Omega$. That is, it satisfies reflexivity ($\theta_1 \sim \theta_1$), symmetry ($\theta_1 \sim \theta_2$ if and only if $\theta_2 \sim \theta_1$), and transitivity (if $\theta_1 \sim \theta_2$ and $\theta_2 \sim \theta_3$, then $\theta_1 \sim \theta_3$). Then, a equivalence class of $\theta_1 \in \Omega$ is defined as $\{\theta | \theta \in \Omega, \theta \sim \theta\}$. We denote this by $[\theta_1]$. Then, we are able to give the definition of model identifiability based on equivalence classes:

**Definition F.1** (Model identifiability based on equivalence relation). *Assume $p_\theta(X)$ is a distribution of some random variable $X$, $\theta$ is its parameter that takes values in some parameter space $\Omega_\theta$, and $sim$ some equivalence relation on $\Omega$ Then, if $p_\theta(X)$ satisfies $p_{\theta_1}(X) = p_{\theta_2}(X) \iff \theta_1 \sim \theta_2 \iff [\theta_1] = [\theta_2], \forall \theta_1, \theta_2 \in \Omega_\theta$, we say that $p_\theta$ is $\sim$ identifiable w.r.t. $\theta$ on $\Omega_\theta$.*

Apparently, definition 2.1 is a special case of definition F.1, where $\sim$ is given by the equality operator, $=$. When the discussion is based on the identifiability under equivalence relation, then it is obvious that all the arguments of Proposition 1, 2, and 3 still holds. Also, the statement of the results needs to adjusted accordingly. For example, in Proposition 1, instead of "the ground truth model parameter $\theta^*$ can be uniquely identified", we now have "the ground truth model parameter $\theta^*$ can be uniquely identified *up to a equivalence relation, $\sim$*".

## G    Subset identifiability (A2) for identifiable VAEs

The GINA model needs satisfy the requirement on model of Proposition 1 or 3, if we wish to use it to fit to the partially observed data and then perform (unbiased) missing data imputation. In order to show that the identifiability result of Proposition 1/3 can be applied to GINA, the key assumption that we need to verify is the local identifiability (**Assumption A2**).

To begin with, in [16], the following theorem on VAE identifiability has been proved:

**Theorem 1.** *Assume we sample data from the model given by $p(X, Z|U) = p_\epsilon(X - f(Z))p_{T,\zeta}(Z|U)$, where $f$ is a multivariate function $f : \mathbb{R}^H \mapsto \mathbb{R}^D$. $p_{T,\zeta}(Z|U)$ is parameterized by exponential family of the form $p_{T,\zeta}(Z|U) \propto \prod_{i=1}^M Q(Z_i) \exp[\sum_{j=1}^K T_{i,j}(Z_i)\zeta_{i,j}(U)]$, where $Q(Z_i)$ is some base measure, $M$ is the dimensionality of the latent variable $Z$, $\mathbf{T}_i(U) = (T_{i,1}, ..., T_{i,K})$ are the sufficient statistics, and $\boldsymbol{\zeta}_i(U) = (\zeta_{i,1}, ..., \zeta_{i,K})$ are the corresponding parameters, depending on $U$. Assume the following holds:*

1. *The set $\{X \in \mathcal{X} | \phi_\epsilon(x) = 0\}$ has zero measure, where $\phi$ is the characteristic function of $p_\epsilon$;*

2. *The multivariate function $f$ is injective;*

3. *$T_{i,j}$ are differentiable a.e., and $(T_{i,j})_{1 \le j \le k}$ are linearly independent on any subset of $\mathcal{X}$ of measure greater than zero;*

4. *There exists $nk + 1$ distinct points $U^0, ..., U^{nk}$, such that the matrix $L = (\boldsymbol{\zeta}(U^1 - U^0), ..., \boldsymbol{\zeta}(U^{nk} - U^0))$ of size $nk$ by $nk$ is invertible.*

*Then, the parameters $(f, T, \zeta)$ are $\sim_A$-identifiable, where $\sim_A$ is the equivalence class defined as (see also Appendix F):*

$$(f, T, \zeta) \sim (\tilde{f}, \tilde{T}, \tilde{\zeta}) \iff \exists \mathbf{A}, \mathbf{c} | \mathbf{T}(f^{-1}(X)) = \mathbf{A}\mathbf{T}(\tilde{f}^{-1}(X)) + \mathbf{c}$$

. *Here,* **A** *is a* $nk$ *by* $nk$ *matrix, and* **c** *is a vector.*

Note that under additional mild assumptions, the **A** in the $\sim_A$ equivalence relation can be further reduced to a permutation matrix. That is, the model parameters can be identified, such that the latent variables differs up to a permutation. This is inconsequential in many applications. We refer to [16] for more discussions on permutation equivalence.

So far, Theorem 1 only discussed the identifiability of $p(X)$ on the *full* variables, $X = X_o \bigcup X_u$. However, in **Assumption A2**, we need the reference model to be (partially) identifiable on a partition $O_s \in \mathcal{A}_\mathcal{I}$, $p_\theta(X_{o_s})$. Naturally, we need additional assumptions on the the injective function $f$, as stated below:

**Assumption A5**  There exists an integer $D_o$, such that $f_O : \mathbb{R}^H \mapsto \mathbb{R}^{|O|}$ is injective for all $O$ that $|O| \geq D_0$. Here, $f_O$ is the entries from the output of $f$, that corresponds to the index set $O$.

**Remark**  Note that, under assumption **A5**, the Assumption **A3** in Section 3 becomes more intuitive: it means that in order to uniquely recover the ground truth parameters, our training data must contain training examples that have more than $D_0$ observed features. This is different from some previous works ([38] for example), where complete case data must be available.

Finally, given these new assumptions, it is easy to show that:

**Corollary 1** (Local identifiability). *Assume that* $p(X, Z|U) = p_\epsilon(X - f(Z))p_{T,\zeta}(Z|U)$ *is the model parameterized according to Theorem 1. Assume that the assumptions in Theorem 1 holds for* $p(X|U)$. *Additionally, assume that* $f$ *satisfies assumption* **A5**.

*Then, consider the subset of variables,* $X_o$. *Then,* $p(X_o|U)$ *is* $\sim_A$-*identifiable on* $(f_O, T, \zeta)$ *for all* $O$ *that satisfies* $|O| \geq D_0$, *where* $f_O$ *is the entries from the output of* $f$, *that corresponds to the index set* $O$.

**Proof**  : it is trivial to see that the assumptions 1, 3, and 4 in Theorem 1 automatically holds regarding $p(X_o|U)$. $f_O$ is injective according to Assumption **A5**. Hence, $p(X_o|U)$ satisfies all the assumptions in Theorem 1, and $p(X_o|U)$ is $\sim_A$-identifiable on $(f_O, T, \zeta)$ for all $O$ that satisfies $|O| \geq D_0$. $\qquad\qquad\square$

**Remark**  In practice, **Assumption A5** is often satisfied. For example, consider the $f$ that is parameterized by the following MLP composite function:

$$f(Z) = h(W \circ g(Z)) \tag{7}$$

, where $g$ is a $D_0$ dimensional, injective multivariate function $g : \mathbb{R}^H \mapsto \mathbb{R}^{D_0}$, $h$ is some activation function $h : \mathbb{R} \mapsto \mathbb{R}$, and $W$ is a injective linear mapping $W : \mathbb{R}^{D_0} \mapsto \mathbb{R}^D$ represented by the matrix $W_{D_0 \times D}$, whose submatrices that consists of $|O| \geq D_0$ arbitrary selected columns are also injective. Note that this assumption for $W$ is not hard to fulfill: a randomly generated matrix (e.g., with element-wise i.i.d. Gaussian prior) satisfies this condition with probability 1. To verify $f_O$ is injective for all $|O| \geq D_0$, notice that $f_O(Z) = h(W_O \circ g(Z))$, where $W_O$ is the output dimensions of $W$ that corresponds to the index set $O$. Since $W$ is injective and $|O| \geq D_0$, we have that $W_O$ is also injective, hence $f_O$ is also injective.

# H  Consistency of estimation for GINA

In [16], a result regarding the consistency of estimation for identifiable VAE. Similarly, it is trivial to show that similar result holds for GINA:

**Theorem 2** (Consistency of estimation). *Assuming that*

1. $q_\phi(Z = z^k|X_o)$ *is expressive enough to contain the true posterior* $p_{\theta,\lambda}(Z|X_o)$, *for all* $X_o$, $\theta$ *and* $\lambda$.

2. *The model in Section 4 is correctly specified, and its parameters are estimated by maximizing* $\mathcal{L}_K(\theta, \lambda, \phi, X_o, R)$ *w.r.t.* $\theta$, $\lambda$, *and* $\phi$.

*Then, under perfect information (infinite samples from data),* $\theta^*$ *and* $\lambda^*$ *is recovered up to* $\sim_A$.

**Proof** Since $q_\phi(Z = z^k|X_o)$ is expressive enough to contain the true posterior, $\mathcal{L}_K(\theta, \lambda, \phi, X_o, R)$ recovers the true likelihood function $\log p_{\theta,\lambda}(X_o, R)$ by simply maximizing $\phi$. Therefore, the problem of maximizing $\mathcal{L}_K(\theta, \lambda, \phi, X_o, R)$ is equivalent to maximum likelihood estimation problem. Therefore, since we assumed that the model is correctly specified, the consistency of MLE trivially implies the consistency of GINA model trained via maximizing $\mathcal{L}_K(\theta, \lambda, \phi, X_o, R)$. $\square$

## I Active question selection

Suppose $X_o$ be the set of observed variables, that represents the correctness of student's response to questions that are presented to them. Then, in the problem of active question selection, we start with $O = \emptyset$, and we would like to decide which variable $X_i$ from $X_U$ to observe/query next, so that it will most likely provide the most valuable information for some target variable of interest, $X_\phi$; Meanwhile, we should while make as few queries as possible. Once we have decided which $X_i$ to observed next, we will make query and add $i$ to $O$. This process is done by maximizing the information reward proposed by [28]:

$$i^* = \arg\max_{i \in U} R(i \mid X_O) := \mathbb{E}_{X_i \sim p(X_i|X_O)} \mathbb{KL}\left[p(X_\phi|X_i, X_O) \,\|\, p(X_\phi|X_O)\right].$$

In the Eedi dataset, as we do not have a specific target variable of interest, it is defined as $X_\phi = X_U$. In this case, $X_\phi$ could be ver high-dimensional, and direct estimation of $\mathbb{KL}\left[p(X_\phi|X_i, X_O) \,\|\, p(X_\phi|X_O)\right].$ could be inefficient. In [28], a fast approximation has been proposed:

$$
\begin{aligned}
R(i \mid X_o) =& \mathbb{E}_{X_i \sim p(X_i|X_o)} D_{KL}\left[p(Z|X_i, X_o)||p(Z|X_o)\right] - \\
& \mathbb{E}_{X_\phi, X_i \sim p(X_\phi, X_i|X_o)} D_{KL}\left[p(Z|X_\phi, X_i, X_o)||p(Z|X_\phi, X_o)\right]. \\
\approx& \mathbb{E}_{X_i \sim \hat{p}(X_i|X_o)} D_{KL}\left[q(Z|X_i, X_o)||q(Z|X_o)\right] - \\
& \mathbb{E}_{X_\phi, X_i \sim \hat{p}(X_\phi, X_i|X_o)} D_{KL}\left[q(Z|X_\phi, X_i, X_o)||q(Z|X_\phi, X_o)\right].
\end{aligned}
$$

In this approximation, all calculation happens in the latent space of the model, hence we can make use of the learned inference net to effceintly estimate $R(i \mid X_o)$.

## J Additional results

### J.1 Imputation results for synthetic datasets

In addition to the data generation samples visualized in Figure 3, we present the imputation results for synthetic datasets in Figure 4. The procedure of generating the imputed samples are as follows. First, each model are trained on the randomly generated, partially observed synthetic dataset described in Section 6.1. Once the models are trained, they are used to impute the missing data in the training set. For each training data, we draw exactly one sample from the (approximate) conditional distribution $p_t heta(X_u|X_o)$. Thus, we have "complete" version of the training set, one for each different model. Finally, we draw the scatter plot for each imputed training set, per dataset and per model. If the model is doing a good job recovering the ground truth distribution $p_\mathcal{D}(X)$ from training set, then its scatter plot should be close to the KDE estimate of the ground truth density of complete data. According to Figure 4, the imputed distribution is similar to the generated distribution in Figure 3.

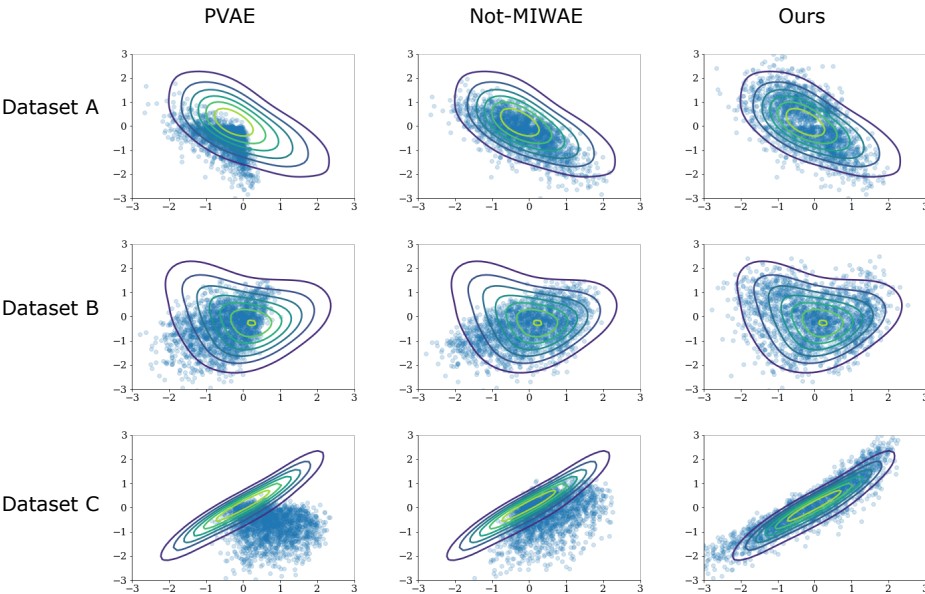

Figure 4: Visualization of imputed $X_2$ and $X_3$ from synthetic experiment. **Row-wise (A-C)** plots for dataset A, B, and C, respectively; **Column-wise:** PVAE imputed samples, Not-MIWAE imputed samples, and GINA imputed samples, respectively. **Contour plot**: kernel density estimate of ground truth density of complete data;