# OpenReview forum: "Identifiable Generative models for Missing Not at Random Data Imputation"
_NeurIPS.cc/2021/Conference — NeurIPS 2021 Poster_

### Official Review · Reviewer_eDY7 · 2021-07-04

**Rating:** 7
**Confidence:** 4

**Summary:**

This paper studies identifiability of generative models for Missing Not at Random (MNAR) data. MNAR is the most general missing mechanism, where the the missingness can depend on both observed and unobserved data. Most prior imputation techniques work under MCAR or MAR assumptions, and there are few models to handle missing data under MNAR. In this respect, model identifiably under MNAR assumption has been mostly overlooked. In fact, the identifiability of models is crucial for learning from MNAR data to uniquely determine the model parameters and have unbiased imputations. This paper provides the sufficient conditions and assumptions to satisfy identifiability of generative models under MNAR. In addition, it proposes a practical deep generative model (GINA) based on VAEs which can satisfy the identifiability conditions.

The contributions are as follows:
* Theoretical analysis of model identifiably under MNAR, which gives sufficient conditions to have identifiable models (uniquely identify the model parameter, matching the ground-truth data generation process).

* A practical algorithm to satisfy these conditions and learn deep generative models under MNAR.

* Experimental studies showing effectiveness of the proposed model on both synthetic and real datasets.

**Limitations And Societal Impact:**

There is no sensitive data in this paper. But, in general, imputation techniques might be used to retrieve the users' information without their consent.

**Main Review:**

### Strengths
- I appreciate the author’s great effort to include both theoretical analysis and the practical model.

- Most imputation papers try to blindly propose a model which gives better results. This paper originates from an important limitation in existing imputation techniques and tries to fill the gap in the state-of-the-art models. Imputation under MNAR and specially model identifiability under MNAR to have unbiased imputations have been mostly overlooked in the prior work. So, I believe this work has significant contribution to the literature for learning from incomplete data.


### Limitations
- I appreciate the experimental studies by the authors on new datasets which are more relevant to MNAR problem. I think the experiments are sufficient. But, as a recommendation for future work, the UCI datasets in [10] or the paper below (e.g. Table 4) could be also used to have a more comprehensive comparison.
    - Yu Gong, et al. "Variational Selective Autoencoder: Learning from Partially-Observed Heterogeneous Data." International Conference on Artificial Intelligence and Statistics, 2021.
    - BTW, this paper can also work under MNAR. But, it does not analyze the model identifiably. It is good to cite this paper and explain the advantage of your work.

### Minor comments
- The paper is somehow hard to read.
    - Sometimes the notation is not conventional and makes it complicated to understand. For example, using $\mathbf{x}$ for random variables is more conventional compared to $X$. I understand that the authors used $X$, $R$, $O$ as sets. But, defining them as sets makes it hard to understand.
    - GINA algorithm is hard to understand. It is not based on the original VAE formulation. So, it is hard to understand and requires reading [14] and [10].
    - The main limitation of the model is discussed in two sentences in Section 7. It was hard for me to understand it. What does the auxiliary variables mean?
    - There are also some typos:
         - The sentence in line 104-105 is not understandable.
         - In line 158, I think $R_{I\O'_l} = 0$ miss $= 0$


### After Rebuttal
I acknowledged I read the authors' response and considered it in the final discussions with the reviewers.


**Time Spent Reviewing:**

8 hours

---

> ### Author Response · Authors · 2021-08-10
> **Response to Reviewer eDY7**
>
> Thank you very much for your extensive and thorough review. We are pleased to see that you appreciate the effort and novelty of our work. We also highly appreciate your feedback for making our work more accessible.
>
>
>
> > **Q1: I appreciate the experimental studies by the authors on new datasets which are more relevant to the MNAR problem. I think the experiments are sufficient. But, as a recommendation for future work, the UCI datasets in [10] or the paper below (e.g. Table 4) could be also used to have a more comprehensive comparison.**
>
> **A1:** We appreciate your suggestions for future work. Firstly, our paper did include standard benchmarks. For example, we have included experiments on Yahoo! R3 in Section 6.2, which is often used in other papers dealing with MNAR,  including the one that is mentioned by the reviewer [A] . Second, we will add (Yu Gong, et al.) into our related work section with an additional discussion. For future work, we will also consider UCI datasets as suggested by the reviewer.
>
> > **Q2: Sometimes the notation is not conventional and makes it complicated to understand.**
>
> **A2:** Thank you so much for the suggestion on the notation system, we will improve the presentation by revisiting some of the notation choices, and recalling them more often in revision and make it more accessible.
>
> > **Q3: It is hard to understand and requires reading [14] and [10].**
>
> **A3:** Thanks for the valuable feedback. In our paper, the basic idea of [10]([A]) is technically covered in Section 2, since their idea is in line with Rubin's formulation and solution of MNAR problems.  In revision, we will make it more clear and add more discussions. On the other hand, regarding [14]([B]), we will add a brief discussion on identifiable VAEs in revision.
>
> > **Q4: The main limitation of the model is discussed in two sentences in Section 7. It was hard for me to understand it. What do the auxiliary variables mean?**
>
> **A4:** Sorry for the confusion. By "auxiliary variables'', we are actually referring to the variable $\mathbf{u}$ in the GINA model (Figure 2), which is just the additional inputs to the prior term, as used in [14]([B]). We will clarify this in revision
>
>
>
> **References**
>
>
> [A] Ipsen, N. B., Mattei, P. A., \& Frellsen, J. (2020). not-MIWAE: Deep generative modelling with missing not at random data. arXiv preprint arXiv:2006.12871.
>
> [B] Khemakhem, I., Kingma, D., Monti, R., \& Hyvarinen, A. (2020, June). Variational autoencoders and nonlinear ica: A unifying framework. In International Conference on Artificial Intelligence and Statistics (pp. 2207-2217). PMLR.

---

### Official Review · Reviewer_NoWR · 2021-07-12

**Rating:** 6
**Confidence:** 5

**Summary:**

This paper proposed a new approach to identify the generative model under missing not at random scenario.
The paper starts with a model on the underlying distribution and then propose assumptions so that the model's parameters are still identifiable even if missing at random is violated.

**Limitations And Societal Impact:**

I have described limitations of the paper in my main review.

**Main Review:**

This paper proposed an interesting theoretical framework on the missing data problem. While I see some novelty in this paper, there are some critical issues that the authors did not address.

Q1. [missing data vs model assumption]
In missing data problems, there are two types of assumptions we need to consider: missing data assumption and model assumption.
This paper starts with a (parametric) model and then impose missing data assumption on it.

While this leads to an identifiable result (under additional assumptions), this identification is not "nonparametric identification".
I think this should be discussed in the paper.


Q2. [Strong assumption (A1-3)]
It seems to me that the main idea that this paper can avoid problems of MNAR is due to assumption (A1-3), which basically imply the problem is similar to the ignorable scenario (MAR).
These assumptions seem to be pretty strong.

Can the authors provide some concrete and simple examples on how these assumptions can be true?
Will the data follows a multivariate Gaussian with some additional constraints make these assumption true?


Q3. [Section 3.2]
This relaxation is rather weak--the one-to-one assumption almost says that the problem remains the same.
What generally would happen is that the data is not generated from a parametric model but a nonparametric model.
In this case, the model we assume will be a best parametric approximation to the target model.
Can we argue that even under this case, our fitted model will be close to the nonparametric model with some bounds on the difference?


Q4. [About GINA]
I was not very comfortable about this part since the neural network itself is a model and is a very complex one.
What are the conditions on f? This should bee clarified.

Also, in appendix G, the authors basically need the generative model to be from the model assumed by GINA.
But it is pretty difficult to believe that a neural net is the actual generative model of the data (neural net can be a good approximation but not the actual generative process)...

Moreover, I don't think that the authors proved GINA satisfy assumption A4 in appendix G.
There needs to be some additional assumptions on the actual generative process when the neural net is not the actual generative model.

Q5. [Model diagnostic]
Given that the authors have assumed a particular model for the generative process, the first thing that we can do is to check if the model assumption makes sense (at least for the observed part).
This should be discussed.
And what should we do if we find out that our assume model does not fit to the data?


**Time Spent Reviewing:**

6

---

> ### Author Response · Authors · 2021-08-10
> **Response to Reviewer NoWR**
>
> Thank you for your valuable review and valuable suggestions to improve.  We are particularly pleased that you appreciate the significance and the clarity of our work. We will try to address each question in the review.
>
> > **Q1: [missing data vs model assumption] ...this identification is not "nonparametric identification". I think this should be discussed in the paper.**
>
> **A1:** It is correct that our identifiability results are "parametric identifiability" not "non-parametric identification''.  This is reflected in our model and missing data assumptions A1 (See our response to Q2 for more discussion on assumptions). In Section 5, we also highlighted that the identifiability under missing data literature can be classified into parametric and non-parametric identification, and we have reviewed related works in both regimes. We will further clarify it in the paper.  While our work focuses on parametric identification, the combination of flexibility of the new deep latent variable model-based solution with additional theoretical insight is novel and enables our method to be used in general real-world applications without additional domain expert input. Apart from the practical method,  as highlighted by other reviewers (DJpx and eDY7), one of the contributions of our paper is to "fill the gap (of identifiability) in the current state-of-the-art models''.
>
> > **Q2: [Strong assumption (A1-3)] They imply the problem is similar to the ignorable scenario (MAR). Can the authors provide some concrete and simple examples?**
>
> **A2:**  Our assumptions and model design explicitly depend on the missing mechanism variable $R$, which is very different from the ignorable scenario of MAR. Moreover, A1-3 are fairly general and being considered mild in related literature. Next, we analyze each assumption.
>
> * Assumption A1:  First, it covers many popular latent variable models in the literature (such as PPCA, VAEs, etc). It also covers a wide range of common missing mechanisms in missing data literature [A, B, C, D]. Therefore, A1 is, in fact, a very common assumption. More importantly, our contribution is beyond A1. For presentation clarity, we presented A1 first (Section 3.1) and relaxed the A1 in the next section (3.2) to cover all different missingness mechanisms, which enable our work to be applied in all different scenarios including all examples apart from (j) in Figure 1.
>
> * Roughly speaking, assumptions A2 and A3 are saying that the problem can be broken down into a set of smaller problems that are easier to solve. That is, the identification problem on subsets of variables $X_{O_s}$, $O_s \in \mathcal{A}_{\mathcal{I}}$ (A2), on which we have a positive probability of obtaining observed data (A3). This does *not* imply that the problem is similar to MAR. One intuitive example is self-masking missing mechanisms (Figure 1 (c)) in combination with certain probabilistic models. For example, the self-masking exponential distribution, and self-masking one-way ANOVA (random effect analysis of variance) [E]. As discussed in [E, Chapter 6.2], these are highly non-MAR examples that will produce estimations that are significantly different from their MAR counterparts. More synthetic examples (self-masking Gaussians, etc.) are also analyzed in our experiments in Section 6.1, which shows that our method is able to solve non-trivial MNAR problems.
>
>
> > **Q3: [Section 3.2] This relaxation is rather weak...**
>
> **A3:** This question will be answered together with Q5 below.
>
> > **Q4: [About GINA] I was not very comfortable about the use of the neural networks...**
>
> **A4:**
>
> * Firstly, we would like to point out that in the related literature regarding identifiability analysis, it is quite common to assume that the actual generative model follows a complicated NN structure (Khemakhem et al., 2020, Liu et al., 2020, Lu et al, 2021, etc). VAEs/NNs becomes the natural choice because we would like to bridge the gap between theoretical insights (that usually considers limited real-world scenarios) and more flexible models (which can be more real-world application friendly).
>
> * Moreover, in the missing data literature, VAEs/NNs are getting more and more popular for practical application [C][L][M][N]. However, identifiability analysis of these models under missing data is still missing. Therefore, as acknowledged by other reviewers (DJpx, REC5, and eDY7), it makes sense to focus on those complex NN models.
>
> * As mentioned in Section 4, the key component of GINA is the identifiable VAE [F]. We assumed that the assumptions of [F] (see Theorem 1 in Appendix G) are satisfied. We will further clarify in the paper.  The only requirement for decoder $f$ is that it should be injective. Although this injective property needs more testing for neural networks, it is a common practice in the identifiability literature [F] [I] [J] to parameterize $f$ using standard neural network architectures (MLPs, Convolutional/Deconvolutional nets, RNNs, etc) and assume that the injective holds. In principle, one can use other more complicated architectures with injectiveness guarantees [H], such as General Incompressible flow Networks (GIN) [G], which will not change our theoretical framework. We will extend this discussion in the paper.
>
> * Finally, our identifiability results in Section 3 are quite general and can be applied to any latent variable models beyond NNs, as long as the assumptions A1-A3 are satisfied. The general form of GINA ($X = f(Z)$ for some $f$) is basically the definition of implicit models [K], which is quite common in scientific geography, ecology, and climate science, where simulators are used to transform stochastic noises $Z$ into observable samples $X$.
>
> > **Q3 \& Q5:**
>
> > **Q3: [Relaxation] The relaxation in Section 3.2 is rather weak;**
>
> > **Q5: [Model diagnostic] Given that the authors have assumed a particular model for the generative process, the first thing that we can do is to check if the model assumption makes sense.**
>
> **A3 \& A5:** Model assumptions can be classified into two categories: i), assumptions regarding graphical structures, and ii), assumptions regarding specific parametric forms.
>
> * **Graphical structure assumptions:** Note that the major goal of the relaxation discussed in Section 3.2 is to deal with the misspecification of *graphical structure assumptions* considering different missingness mechanisms. Practically, we have shown that (line 225-231) we can use the model structure shown in Figure 2 ( designed to be consistent with Figure 1(h)) to cover general MNAR situation (for example, all situation in Figure 1 apart from (j)). Given this relaxation result in Section 3.2, we can avoid testing the assumptions of the underlying data generation graphical structure.
>
> * **Assumptions regarding parametric forms:** We would like to comment that as in the whole field of machine learning, many models have assumptions regarding their parametric forms, and not all these assumptions can be tested. Unfortunately, we are not able to analyze *all possible data generating processes* that satisfies assumptions A1-A3.  Instead, we have to pick one specific parameterization assumption for the data generating process. Due to reasons described in our response to Q4, we choose to focus on VAE/NNs due to its flexibility and practical performance. The superior performance GINA in our evaluation  confirms that our parametric form is reasonable. Given the scope of our paper (analysis of identifiability of the novel *deep generative model* for missing value imputation under MNAR), we believe that this is already a shred of strong evidence.
>
>
> **References**
>
> [A] Rubin, D. B. (1976). Inference and missing data. Biometrika, 63(3), 581-592.
>
> [B] Mohan, K., Pearl, J., \& Jin, T. (2013) Graphical models for inference with missing data. Advances in neural information processing systems, 26:1277–1285
>
> [C] Ipsen, N. B., Mattei, P. A., \& Frellsen, J. (2020). not-MIWAE: Deep generative modelling with missing not at random data. arXiv preprint arXiv:2006.12871.
>
> [D] Sportisse, A., Boyer, C., \& Josses, J. (2020). Estimation and imputation in probabilistic principal component analysis with missing not at random data. Advances in Neural Information Processing Systems, 33.
>
> [E] Little, R. J., \& Rubin, D. B. (2019). Statistical analysis with missing data (Vol. 793). John Wiley \& Sons.
>
> [F] Khemakhem, I., Kingma, D., Monti, R., \& Hyvarinen, A. (2020, June). Variational autoencoders and nonlinear ica: A unifying framework. In International Conference on Artificial Intelligence and Statistics (pp. 2207-2217). PMLR.
>
> [G] Sorrenson, P., Rother, C., \& Köthe, U. (2020). Disentanglement by nonlinear ica with general incompressible-flow networks (gin). arXiv preprint arXiv:2001.04872.
>
> [H] Zhou, D., \& Wei, X. X. (2020). Learning identifiable and interpretable latent models of high-dimensional neural activity using pi-VAE. arXiv preprint arXiv:2011.04798.
>
> [I] Liu, C., Sun, X., Wang, J., Tang, H., Li, T., Qin, T., ... \& Liu, T. Y. (2020). Learning causal semantic representation for out-of-distribution prediction. arXiv preprint arXiv:2011.01681.
>
> [J] Lu, C., Wu, Y., Hernández-Lobato, J. M., \& Schölkopf, B. (2021). Nonlinear invariant risk minimization: A causal approach. arXiv preprint arXiv:2102.12353.
>
> [K] Li, Y., \& Turner, R. E. (2017). Gradient estimators for implicit models. arXiv preprint arXiv:1705.07107.
>
> [L] Mattei, P. A., \& Frellsen, J. (2019, May). MIWAE: Deep generative modelling and imputation of incomplete data sets. In International Conference on Machine Learning (pp. 4413-4423). PMLR.
>
> [M] Nazabal, A., Olmos, P. M., Ghahramani, Z., \& Valera, I. (2020). Handling incomplete heterogeneous data using vaes. Pattern Recognition, 107, 107501.
>
> [N] Ma, C., Tschiatschek, S., Palla, K., Hernández-Lobato, J. M., Nowozin, S., \& Zhang, C. (2018). Eddi: Efficient dynamic discovery of high-value information with partial vae. arXiv preprint arXiv:1809.11142.

---

### Official Review · Reviewer_REC5 · 2021-07-13

**Rating:** 8
**Confidence:** 5

**Summary:**

The article is concerned with model identifiability in Missing Not At Random (MNAR) data setting, with a particular focus on deep generative models. The authors introduce a unified view of model identifiability under MNAR, and derive sufficient conditions for identifiability in several MNAR scenarios, which highlight new important concepts. They also introduce a new imputation method for MNAR data based on identifiable VAE models, which is submitted as a python code, and extensively tested on synthetic and real data sets against state of the art methods for MNAR imputation.

**Limitations And Societal Impact:**

The authors adequately addressed the limitations and potential negative societal impact of their work

**Main Review:**

Identifiability of MNAR data models is an important topic, as MNAR data is a major problem in applied problem which often impairs analysis, and few methods have been developed to overcome this. Even fewer existing work provide some theoretical understanding in such settings. In addition, existing results on identifiability usually lead to computationally costly imputation methods which are not easily generalized. Thus, I think this article is important and quite remarkable, as it provides original theoretical results, along with an efficient software and exhaustive numerical simulations, as well as an actual application to a large scale education data set; I believe the method could have an important impact in applications. I have a few questions for the authors
- Concerning Assumptions 2 and 3, is there a dependence somewhere in your results in the number of partitions S and L and in their size ?
- Do you believe these identifiability results could pave the way for estimation consistency in MNAR settings ?
- What kind of architecture did you use in practice for the VAE in GINA ? Did you use different ones for each problem ? How robust do you think the procedure is to the choice of architecture ?

I have read the authors' rebuttal, which provide relevant answers to my questions. The authors added simulations to empirically evaluate the robustness of the method to the networks' architecture.

**Time Spent Reviewing:**

2

---

> ### Author Response · Authors · 2021-08-10
> **Response to Reviewer REC5**
>
> Thank you very much for your positive opinions about our work. We are pleased to see that you appreciate the technical novelty and the practical impact of our work. In the following, we will try to address each question in the review.
>
>
>
> > **Q1: Do you believe these identifiability results could pave the way for estimation consistency in MNAR settings**
>
> **A1:** We believe so. Since the identifiable VAE [A] is a building block of our GINA model, we can follow the similar reasoning of the consistency analysis in [A] and derive similar results under MNAR. Of course, we may need to introduce additional assumptions similar to [A] to show estimation consistency.  We will extend the discussion in revision.
>
> > **Q2: What kind of architecture did you use in practice for the VAE in GINA? How robust do you think the procedure is to the choice of architecture?**
>
> **A2:** Generally speaking, we only used simple MLPs for the decoder part of both GINA and VAE. For example, we used a two-layer MLP with 20 latent dimensions for the Yahoo dataset, and a three-later MLP with 50 latent dimensions for the Eedi dataset which is resulted from grid search regarding depth and width using the validation set as shown in Appendix B. For encoders, we used zero imputing for simple synthetic data experiment in 6.1 as in [B] and point net structure as in [C] which is more efficient in high dimensional data for real-world data sets
> (6.2, 6.3).
>
> To further showcase the robustness of the GINA structure, we consider the Yahoo R3 experiment as done in Section 6.2. In the table below, we compute the imputation error of GINA against a wide range of different GINA decoder structures. We set the encoder structure to be symmetric to the decoder. For example, if the decoder has structure 20-50-D, then the encoder structure is D-50-20. As we can observe, as the GINA network gets bigger, the change of imputation error is not very drastic, and the optimal imputation error is achieved with a 20-50-100-200-D structure.
>
> | Structure | 20-10-D | 20-50-D | 20-50-50-D | 20-50-100-D | 20-50-100-200-D | 100-100-100-100-D |
> |-----------|:---------:|:---------:|:------------:|:-------------:|:-----------------:|:-------------------:|
> | Mean MSE | 1.052   | 1.030   | 1.043      | 1.032       | **1.027**           | 1.051             |
>
> > **Q3: Concerning Assumptions 2 and 3, is there a dependence somewhere in your results in the number of partitions S and L and in their size ?**
>
> **A3:** This may be a misunderstanding. Firstly, the values $S$ and $L$ are *not* hyperparameters. Assumptions 2 and 3 are simply statements regarding the existence (but not uniqueness) of $S$ and $L$. Given a model and a missing mechanism, there often exists multiple values of $S$ and $L$ that satisfy Assumptions 2 and 3. Thus, the specific value they take, nor the *exact number* of all possible partitions (which is also hard to calculate) is irrelevant for the model or the analysis as far as they exist.
> Therefore, we believe that it is not needed to analyze the dependence of empirical results to $S$ and $L$, as well as the number of possible partitions.
>
> **References**
>
> [A] Khemakhem, I., Kingma, D., Monti, R., \& Hyvarinen, A. (2020, June). Variational autoencoders and nonlinear ica: A unifying framework. In International Conference on Artificial Intelligence and Statistics (pp. 2207-2217). PMLR.
>
> [B] Ipsen, N. B., Mattei, P. A., \& Frellsen, J. (2020). not-MIWAE: Deep generative modelling with missing not at random data. arXiv preprint arXiv:2006.12871.
>
> [C] Ma, C., Tschiatschek, S., Palla, K., Hernández-Lobato, J. M., Nowozin, S., \& Zhang, C. (2018). Eddi: Efficient dynamic discovery of high-value information with partial vae. arXiv preprint arXiv:1809.11142.

---

### Official Review · Reviewer_DJpx · 2021-07-16

**Rating:** 7
**Confidence:** 3

**Summary:**

This paper is about identifiability of the data generating process and the missing value mechanism in a missing-not-at-random setting. The paper provides an analysis of identifiability of generative models and propose a deep generative model with identifilibity guarantees under some assumptions. There are existing methods for handling MNAR without identifiability guarantees and there are theoretical analyses on identifiability without practical scalable algorithms. The paper aims to bridge this gap.
A VAE based model for imputation is proposed and compared to two other baselines.  The proposed method models the data and missing mechanism jointly, and in order to ensure identifiability the proposed method needs auxiliary variables that are always observed.

**Limitations And Societal Impact:**

The authors have adequately addressed the limitations and potential negative societal impact of their work.

**Main Review:**

The paper is well written and clearly presented. The focus on MNAR is important as it has been much overlooked compared to how prevalent it is in real world data and the focus on identifiability is useful to further the use of models handling MNAR data.
It nicely address the challenges of MNAR and identifiability and builds up its case coherently. Identifiability is first analyzed under the correct graphical model, then relaxed to cover data model mismatch. The model assumptions are clearly stated as well as the assumptions about the data generating process.

Identifiable models makes sure that the model parameters can be uniquely determined, that the parameter estimates are unbiased.
- In line 38 and 120 this is taken to also be the case for imputations from such models, that identifiability is needed to achieve unbiased imputations. Is this well established? could you provide any references on this? Of course if you indeed get unbiased estimates of the ground truth parameters, imputations will be unbiased, but couldn't two (non-indentifiable) models with different parameter settings in principle give the same (unbiased) imputations? Or is it that non-identifiability CAN lead to biased imputations?

The experiments are few, but nice. Figure 3 is informative as a model evalution using another metric than imputation MSE.
- Is it clear that the difference between GINA and not-MIWAE is due to identifiability and not just plain old model misspecification? that is, the missing mechanism is based on the latent variable, GINA uses the latent variable as input to the missing model while the not-MIWAE does not. Could the problem of model misspecification be separated from the issue of non-identifiability in some way in this experiment?

Minor comments:
- Eq (1) is referred to as the selection model factorization, but this is just the joint (which could be factorized as selection model or pattern mixture), maybe update text or equation.
- Line 177: should it be proposition 1 instead of proposition 2?
- Maybe this work [1] on MNAR is also of interest to you.
- Appendix B), I think you mean the the not-MIWAE is described by figure 1 d), not figure 1 a) (and the PVAE is described by figure 1 a)).

[1] Ghalebikesabi, Sahra, et al. "Deep Generative Missingness Pattern-Set Mixture Models." International Conference on Artificial Intelligence and Statistics. PMLR, 2021.

**Time Spent Reviewing:**

2

---

> ### Author Response · Authors · 2021-08-10
> **Response to Reviewer DJpx**
>
>   Thank you for your encouraging review and valuable suggestions to improve.  We are particularly pleased that you appreciate the significance of our work and the clarity of the presentation. In the following, we will try to address each question in the review.
>
>
>
> > **Q1: In line 38 and 120, is identifiability needed to achieve unbiased imputations.**
>
> **A1:** In our paper (line 38), we have stated that "missing value imputation based on such parameter estimation *could* be biased'' when the model is non-identifiable. Therefore, to answer the reviewer's question, identifiability is indeed the sufficient (but not necessary) condition for unbiased imputations. This fact is theoretically trivial (as mentioned in lines 176-179), and is empirically observed in related works such as [A].
>
> > **Q2: Is it clear that the difference between GINA and not-MIWAE is due to identifiability and not just plain old model misspecification?**
>
> **A2:** Non-identifiability and model misspecification are two problems of not-MIWAE as discussed in Section 1 of the paper, thus, you are correct. This is exactly why we designed Experiment 6.1, where not-MIWAE does not have the misspecification problem for dataset A, but for B and C.  In experiment 6.1 Dataset A, the ground truth missing mechanism does not depend on the latent variable model, in this case, the less satisfying performance is due to non-identifiability. In Dataset B and C, MIWAE additionally suffers from model misspecification, which leads to even worse performance. We will make this more clear in the paper.
>
> > **Q3: Minor issues**
>
> **A3:** Thank you for noticing the typos, we will change them in the revised version, including updating the texts for Eq (1), fixing typos in line 177 and Appendix B, as well as adding the discussion regarding the paper (Ghalebikesabi, Sahra, et al) mentioned by the reviewer.
>
> **References**
>
> [A] Sportisse, A., Boyer, C., \& Josses, J. (2020). Estimation and imputation in probabilistic principal component analysis with missing not at random data. Advances in Neural Information Processing Systems, 33.

---

### Decision · Program_Chairs · 2021-09-27

**Decision:**

Accept (Poster)

**Comment:**


The paper is on imputing missing variables when they are missing not at random (MNAR). All reviewers agree that the paper makes a valuable contribution on a topic that is important but often overlooked. This is a clear accept.

In the camera-ready version, please incorporate the reviewers' feedback. Moreover, during discussion, reviewers brought up the idea of illustrating assumptions A1-A3 on a simple example, which I would indeed recommend to do (e.g in the appendix).